# *Cul3* and *insomniac* are required for rapid ubiquitination of postsynaptic targets and retrograde homeostatic signaling

Koto Kikuma [1], Xiling Li[1], Sarah Perry[1], Qiuling Li[2], Pragya Goel [1], Catherine Chen[1], Daniel Kim[1], Nicholas Stavropoulos[2] & Dion Dickman [1]

At the *Drosophila* neuromuscular junction, inhibition of postsynaptic glutamate receptors activates retrograde signaling that precisely increases presynaptic neurotransmitter release to restore baseline synaptic strength. However, the nature of the underlying postsynaptic induction process remains enigmatic. Here, we design a forward genetic screen to discover factors in the postsynaptic compartment necessary to generate retrograde homeostatic signaling. This approach identified *insomniac* (*inc*), a putative adaptor for the Cullin-3 (Cul3) ubiquitin ligase complex, which together with Cul3 is essential for normal sleep regulation. Interestingly, we find that Inc and Cul3 rapidly accumulate at postsynaptic compartments following acute receptor inhibition and are required for a local increase in mono-ubiquitination. Finally, we show that Peflin, a $Ca^{2+}$-regulated Cul3 co-adaptor, is necessary for homeostatic communication, suggesting a relationship between $Ca^{2+}$ signaling and control of Cul3/Inc activity in the postsynaptic compartment. Our study suggests that Cul3/Inc-dependent mono-ubiquitination, compartmentalized at postsynaptic densities, gates retrograde signaling and provides an intriguing molecular link between the control of sleep and homeostatic plasticity at synapses.

[1] Department of Neurobiology, University of Southern California, Los Angeles, CA 90089, USA. [2] Neuroscience Institute, Department of Neuroscience and Physiology, NYU School of Medicine, New York, NY 10016, USA. Correspondence and requests for materials should be addressed to D.D. (email: dickman@usc.edu)

To maintain stable synaptic activity in the face of stress during development, experience, and disease, the nervous system is endowed with robust forms of adaptive plasticity that homeostatically adjust synaptic strength[1,2]. This homeostatic control of synaptic plasticity is conserved from invertebrates to humans[1,3], and dysfunction in this process is linked to complex neural diseases, including Parkinson's, schizophrenia, Fragile X Syndrome, and autism spectrum disorder[4,5]. Homeostatic adaptations at synapses are expressed through coordinated modulations in the efficacy of presynaptic neurotransmitter release and/or postsynaptic receptor abundance[2,6]. Although it is apparent that a dialog involving both anterograde and retrograde trans-synaptic signaling serves to initiate, maintain, and integrate the homeostatic tuning of synaptic strength, the molecular nature of this communication is largely unknown.

The *Drosophila* neuromuscular junction (NMJ) is an established model system to interrogate the genes and mechanisms that mediate the homeostatic stabilization of synaptic strength. At this glutamatergic synapse, genetic loss or pharmacological inhibition of postsynaptic receptors initiates a retrograde signaling system that instructs a compensatory increase in presynaptic neurotransmitter release that restores baseline levels of synaptic strength[3,7], a process referred to as *presynaptic homeostatic potentiation* (PHP). Forward genetic screens in this system have proven to be a powerful approach to identify genes necessary for the expression of PHP[3,8,9]. Work over the past decade has revealed that a rapid increase in both presynaptic $Ca^{2+}$ influx and the size of the readily releasable vesicle pool are necessary to homeostatically enhance presynaptic neurotransmitter release during PHP[10–13]. Furthermore, candidate molecules involved in retrograde signaling have been proposed[14,15]. However, despite these significant insights, forward genetic screens have failed to shed light on the postsynaptic mechanisms that induce retrograde signaling, a process that remains enigmatic[16–18].

Little is known about the signal transduction system in the postsynaptic compartment that mediates retrograde homeostatic communication. It is clear that pharmacological blockade or genetic loss of GluRIIA-containing receptors initiates retrograde PHP signaling. Perturbation of these receptors leads to reduced levels of active (phosphorylated) $Ca^{2+}$/calmodulin-dependent protein kinase II (CaMKII)[13,17,19,20]. However, inhibition of postsynaptic CaMKII activity alone is not sufficient to induce PHP expression[19], suggesting that additional signaling in the postsynaptic compartment is required to generate instructive retrograde communication. Furthermore, rapid PHP signaling induced by pharmacological receptor blockade does not require new protein synthesis[17,21,22]. Finally, PHP-related reductions in CaMKII signaling is compartmentalized at postsynaptic densities, where PHP can be expressed with specificity at subsets of synapses within a single motor neuron[13,20,23], suggesting that retrograde communication happens locally between individual pre- and postsynaptic dyads. Although a role for postsynaptic PI3-cll kinase in PHP was recently proposed[18], it is unclear how this signaling is connected to localized glutamate receptor perturbation, compartmentalized changes in CaMKII activity, or retrograde communication to specific presynaptic release sites. Altogether, these data suggest that translation-independent signaling systems are compartmentalized at postsynaptic densities and function in addition to CaMKII to ultimately drive retrograde homeostatic communication to specific presynaptic release sites.

To gain insight into the mechanisms underlying PHP induction, we have designed complementary forward genetic screens to identify genes that specifically function in the postsynaptic compartment to enable retrograde homeostatic signaling. This approach discovered a single gene, *insomniac* (*inc*). *inc* encodes a putative adaptor for the Cullin-3 (Cul3) E3 ubiquitin ligase complex and is necessary for normal sleep behavior[24,25]. Our findings suggest that rapid and compartmentalized mono-ubiquitination at postsynaptic densities is a key inductive event necessary for trans-synaptic homeostatic signaling.

## Results

**Electrophysiology-based forward genetic screens identify *inc*.** We first generated a list of ~800 neural and synaptic genes to screen mutants for defects in the ability to express PHP. A substantial portion of these were gleaned from studies linking genes and transcripts to schizophrenia, intellectual disability, autism, and Fragile X Syndrome (see Methods for more details). We hypothesized that transcripts targeted by the Fragile X Mental Retardation Protein (FMRP) in particular might provide a rich source to assess for postsynaptic roles in homeostatic synaptic signaling. First, previous studies have established intriguing links between homeostatic plasticity and complex neurological and neuropsychiatric diseases[4,5]. Second, FMRP itself has important roles at postsynaptic densities[26] and has been implicated in homeostatic signaling at mammalian synapses[27,28]. Third, recent work has demonstrated that modulation of postsynaptic translation is necessary to sustain PHP expression over chronic time scales[16,17,29]. From the initial list of ~800 genes, we established a collection of *Drosophila* homologs (see Methods and Supplementary Data 1) and obtained 197 mutants and 341 RNAi lines to screen for potential defects in PHP expression (Fig. 1a, d). We removed any mutants or RNAi lines that failed to survive to third-instar larval stages or failed to exhibit the expected electrophysiological phenotype for the subset of genes with known roles in promoting basal neurotransmission. This led to a final list of 124 mutants and 249 RNAi lines to screen (Supplementary Data 1).

We used two distinct ways to screen mutants and RNAi lines for their effects on PHP expression. To screen the 124 mutants, we leveraged an established approach that utilizes a rapid pharmacological assay to assess PHP[8,9]. In this assay, application of the postsynaptic glutamate receptor antagonist philanthotoxin (PhTx) inhibits miniature neurotransmission, but synaptic strength (evoked amplitude) remains similar to baseline values because of a homeostatic increase in presynaptic neurotransmitter release (quantal content). For each mutant, we quantified synaptic strength following 10 min incubation in PhTx (Fig. 1b). This led to the identification of eleven potential PHP mutants with reduced excitatory postsynaptic potential (EPSP) amplitude at least two standard deviations below the mean after PhTx application (< 22 mV). This reduction in synaptic strength could be due to either reduced baseline transmission or a failure to express PHP. Therefore, baseline transmission was assessed in these mutants by recording in the absence of PhTx; five mutants with reduced baseline neurotransmission but persistent PHP expression were identified and not studied further (Supplementary Data 1). The remaining six mutants represent genes necessary to express PHP (Fig. 1a, b), including the active zone component *fife*, which was recently shown to be necessary for PHP expression[30]. It is important to note, however, that because the majority of the lines screened have not been previously characterized and do not unambiguously eliminate gene function, we cannot conclude that the negative candidates from our screen are not involved in synaptic transmission or PHP expression.

In parallel, we assessed PHP in the 249 RNAi lines using an established stock that drives the RNAi transgene in both neurons and muscle. In addition, postsynaptic glutamate receptor expression is also reduced in this line through RNAi-mediated knock-down of the *GluRIII* receptor transcript[31]. After crossing each RNAi line to this stock, we quantified electrophysiological

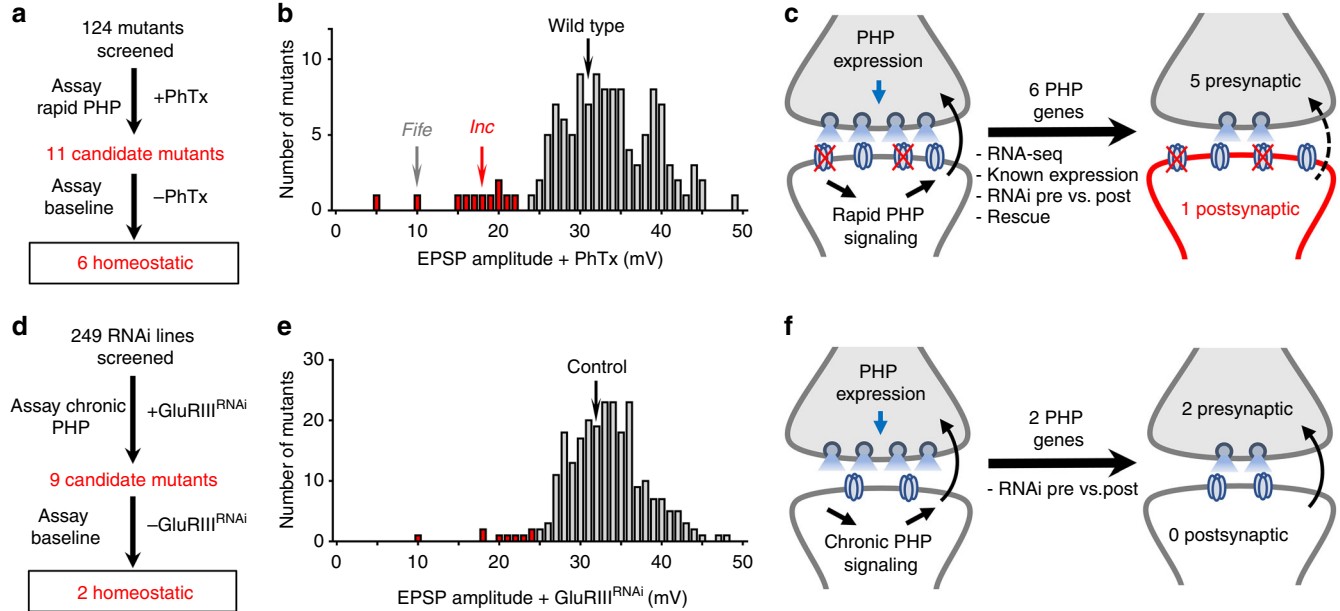

**Fig. 1** Forward genetic screens identify six genes necessary for PHP in distinct synaptic compartments. **a** and **d** Electrophysiology-based forward genetic screening strategy and outcomes for the PhTx (**a**) and *GluRIII* knock down (**d**) approaches. **b** and **e** Average EPSP amplitudes of each mutant or RNAi line screened following PhTx application (**b**) or *GluRIII* knock down (**e**). In wild-type controls, inhibition of glutamate receptors results in reduced mEPSP amplitude, as expected. However, EPSP amplitude remains similar to baseline values due to a homeostatic increase in presynaptic neurotransmitter release (quantal content). Highlighted in red are all mutants that showed EPSP values > two standard deviations below controls. **c** and **f** Schematic illustrating the determination of pre- and postsynaptic functions for the positive hits from the screens. See Supplementary Data 1 for detailed information about all genes screened and additional data

recordings and identified nine genes that were putatively necessary for PHP expression (Fig. 1d). To determine baseline synaptic strength in these RNAi lines, we expressed each in neurons and muscle in the absence of *GluRIII* knock down. Of these nine lines, seven exhibited a significant decrease in EPSP amplitude after crossing to the control stock, suggesting reduced baseline transmission (Supplementary Data 1). In contrast, two RNAi lines displayed normal baseline synaptic strength, indicating they were specifically necessary for PHP expression (Fig. 1d, e). Importantly, these two genes targeted by RNAi knock down were also identified in the PhTx screen, validating this complementary screening strategy. Altogether, these two screens identified five genes whose requirement for PHP has not been previously described.

If a gene functioned in the presynaptic neuron, this would imply that it was involved in increasing neurotransmitter release characteristic of PHP expression, while a postsynaptic function in the muscle would suggest a requirement in the induction of retrograde PHP signaling. We therefore used several strategies to determine in which synaptic compartment each gene was required. For each of the six genes identified in our screen, we assessed RNA-seq expression profiles at the larval NMJ[16], known expression patterns, genetic rescue and/or tissue-specific RNAi knock down (Fig. 1c, f). This analysis revealed five genes necessary in the presynaptic neuron, including *fife*, and only a single gene, *insomniac* (*inc*), that functions in the postsynaptic cell (Fig. 1c, f). Given that the postsynaptic mechanisms that drive the induction of PHP are poorly understood, we focused on characterizing the role of *inc* in PHP signaling.

**inc is required in the postsynaptic muscle for PHP expression.** To further investigate the role of *inc* in PHP, we first generated new null alleles using CRISPR/Cas-9 genome editing technology. We obtained two independent mutations in the *inc* locus causing premature stop codons (Fig. 2a), alleles we named $inc^{kk3}$ and

$inc^{kk4}$. We confirmed that both alleles are protein nulls by immunoblot analysis with an anti-Inc antibody (Fig. 2b). Behavioral analysis demonstrated that both $inc^{kk}$ mutants exhibit severely shortened sleep, similar to previously described *inc* null alleles[24] (Supplementary Fig. 1).

Next, we characterized synaptic physiology in *inc* mutants using two-electrode voltage clamp recordings. We first confirmed that baseline synaptic transmission and postsynaptic glutamate receptor levels were largely unperturbed by the loss of *inc* (Fig. 2c and Supplementary Figs. 1 and 2). However, while PhTx application reduced miniature excitatory postsynaptic current (mEPSC) amplitudes in both wild-type and *inc* mutants, no homeostatic increase in presynaptic neurotransmitter release was observed in *inc* mutants, resulting in reduced excitatory postsynaptic current (EPSC) amplitude (Fig. 2c, d). Similar results were found for $inc^{kk3}/inc^{Df}$ and $inc^{kk4}$ mutants (Fig. 2d and Supplementary Table 2), while heterozygous mutants exhibited wild-type responses (Fig. 3c, d), indicating that these mutants are recessive. In addition, *inc* mutants failed to express PHP over chronic time scales when combined with *GluRIIA* mutations (Supplementary Fig. 2c, d). Thus, *inc* is necessary for the expression of PHP over both rapid and chronic time scales but is dispensable for baseline neurotransmission.

We used an *inc-Gal4* transgene[24] to express a GFP reporter and observed the GFP signal, representing *inc* expression, in both presynaptic motor neurons and the postsynaptic musculature (Fig. 2e), as previously described[32]. If *inc* were required in the neuron for PHP expression, this would indicate a function in augmenting presynaptic neurotransmitter release. In contrast, if *inc* were required in the muscle, this would suggest a role in postsynaptic retrograde communication. Unfortunately, *inc* RNAi approaches were ineffective as PHP remained intact following expression of *inc* RNAi in both pre- or postsynaptic compartments (Supplementary Data 1), consistent with the RNAi only moderately reducing *inc* expression (see below). Therefore, to

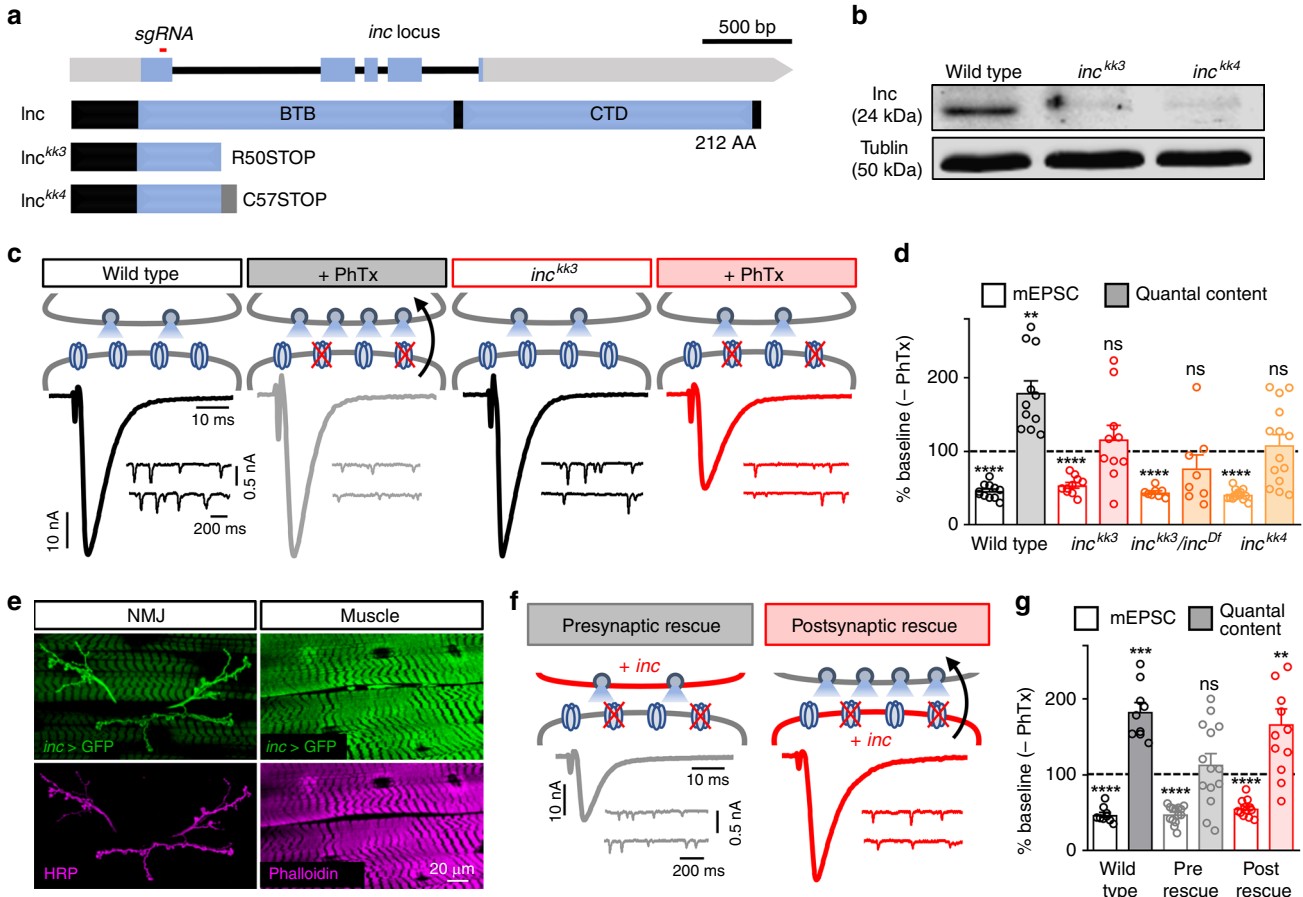

**Fig. 2** *inc* is required in the postsynaptic compartment to drive retrograde PHP signaling. **a** Schematic of the *Drosophila inc* locus, with the region targeted by the single guide RNA to generate the $inc^{kk3}$ and $inc^{kk4}$ alleles shown. (Bottom) Structure of Inc and the predicted structure of the $inc^{kk3}$ and $inc^{kk4}$ mutant alleles. **b** Anti-Inc immunoblot analysis from whole-adult lysates confirms that both $inc^{kk3}$ and $inc^{kk4}$ are protein null alleles. **c** Rapid expression of PHP requires *inc*. Schematic and representative EPSC and mEPSC traces for wild-type ($w^{1118}$) and $inc^{kk3}$ mutants before and after PhTx application. While mEPSC amplitude is reduced after PhTx application, as expected, $inc^{kk3}$ mutants fail to homeostatically increase presynaptic neurotransmitter release, resulting in reduced EPSC amplitudes. **d** Quantification of mEPSC amplitude and quantal content values following PhTx application normalized to baseline values (−PhTx) are shown for the indicated genotypes (−PhTx: wild type, $n = 17$; $inc^{kk3}$, $n = 16$; $inc^{kk3}/inc^{Df}$, $n = 10$; $inc^{kk4}$, $n = 14$; + PhTx: wild type, $n = 11$; $inc^{kk3}$, $n = 10$; $inc^{kk3}/inc^{Df}$, $n = 8$; $inc^{kk4}$, $n = 14$). **e** Representative muscle 6/7 NMJ images of GFP expression driven by the *inc* promoter (*inc-Gal4;UAS-CD4-td-eGFP/ +* ). Anti-HRP (neuronal membrane marker) and anti-phalloidin (actin marker) are shown. *inc* is expressed in both presynaptic motor neurons and postsynaptic muscles. **f** Schematic and representative EPSC and mEPSC traces in which *UAS-smFP-inc* is expressed in motor neurons in *inc* mutant backgrounds (presynaptic rescue: $inc^{kk3};OK371$-*Gal4/UAS-smFP-inc*) or muscle (postsynaptic rescue: $inc^{kk3};UAS$-*smFP-inc/ +;MHC-Gal4/ +* ) following PhTx application. Postsynaptic expression of *inc* fully restores PHP expression, while PHP fails in the presynaptic rescue condition. **g** Quantification of mEPSC and quantal content values in the indicated genotypes relative to baseline (−PhTx: wild type, $n = 10$; presynaptic rescue, $n = 16$; postsynaptic rescue, $n = 12$; + PhTx: wild type, $n = 9$; presynaptic rescue, $n = 14$; postsynaptic rescue, $n = 10$). Asterisks indicate statistical significance using a Student's *t*-test: (**) $p < 0.01$; (***) $p < 0.001$; (****) $p < 0.0001$, (ns) not significant. Error bars indicate ± SEM. *n* values indicate biologically independent cells. Additional statistical information and absolute values for normalized data can be found in Supplementary Table 2. Source data are provided as a Source Data file

determine in which compartment *inc* expression was required for PHP, we performed a tissue-specific rescue experiment using a *UAS* transgene expressing *inc* fused to a *spaghetti monster* Fluorescent Protein (smFP) containing ten copies of the Flag epitope[33] (*UAS-smFP-inc*). Consistent with the notion that smFP-Inc does not antagonize endogenous Inc, overexpression of this transgene had no impact on baseline synaptic transmission or PHP expression (Supplementary Fig. 3). Expression of this transgene with *inc-Gal4* also rescued the sleep deficits in *inc* mutants (Supplementary Fig. 1), suggesting that smFP-Inc recapitulates Inc function. Importantly, PHP expression was fully restored in *inc* mutants when this transgene was expressed specifically in the postsynaptic muscle, but not when expressed in

the presynaptic neuron (Fig. 2f, g). These experiments indicate that *inc* function in the postsynaptic muscle is necessary to enable retrograde PHP signaling.

**Inc and Cul3 transduce retrograde PHP signaling.** *inc* was originally identified in a forward genetic screen for *Drosophila* mutants with reduced sleep[24,25]. *inc* encodes a highly conserved protein with homology to the Bric-à-brac, Tramtrack, and Broad/Pox virus zinc finger (BTB/POZ) superfamily, which includes adaptors for the Cul3 E3 ubiquitin ligase complex. Inc physically interacts with Cul3, and Cul3 is similarly required for normal sleep, suggesting that Inc is a substrate adaptor for the Cul3

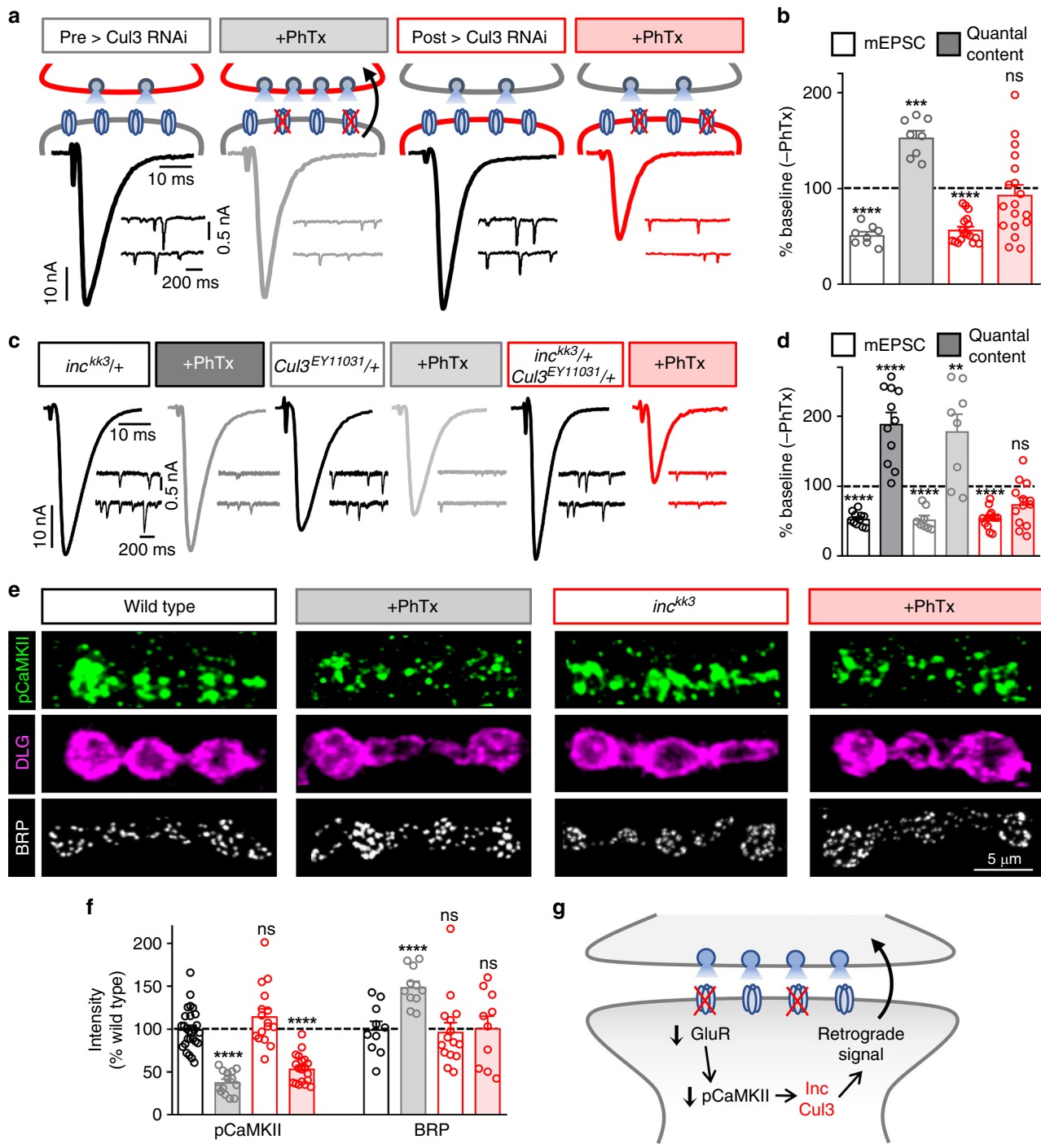

complex[24,25]. Consistent with such a mechanism, we observed that knock-down of *Cul3* in muscle, but not in neurons, disrupted the expression of PHP after PhTx application (Fig. 3a, b), and we also found that *inc* and *Cul3* genetically interact, as indicated by a block in PHP expression in $inc^{kk3}/+$ ;$Cul3^{EY11031}/+$ transheterozygotes (Fig. 3c, d). These findings suggest that Cul3 works with Inc in the postsynaptic compartment to drive retrograde PHP signaling.

Next, we sought to define at what point Inc is required in postsynaptic PHP transduction. First, we assessed whether postsynaptic glutamate receptors, key components that initiate PHP signaling, are altered in *inc* mutants. However, we found no significant difference in glutamate receptor signal intensity or

localization (Supplementary Fig. 2a, b). Next, we examined the compartmentalized reduction in CaMKII activity, thought to be a key inductive event during retrograde PHP signaling. Reduced levels of active (phosphorylated) pCaMKII immunofluorescence intensity at the postsynaptic NMJ are observed following loss or pharmacological blockade of glutamate receptors[17,20,23], and inhibiting this reduction by postsynaptic expression of a constitutively active form of CaMKII (CaMKII[T287D]) occludes chronic PHP expression[19,23]. As the ubiquitination of Inc substrates could trigger their proteolysis, we considered whether Inc might degrade active CaMKII following glutamate receptor perturbation. If so, *inc* mutants might fail to reduce pCaMKII abundance at postsynaptic densities following PhTx application.

**Fig. 3** *inc* and *Cul3* function downstream of CaMKII but upstream of retrograde PHP signaling. **a** Schematic and representative EPSC and mEPSC traces from neuronal *Cul3* knock down (pre > Cul3 RNAi: UAS-Cul3 RNAi[11861R-2]/OK371-Gal4) and muscle *Cul3* knock down (post > Cul3 RNAi: UAS-Cul3 RNAi[11861R-2]/ + ; MHC-Gal4/ + ) before and after PhTx application. post > Cul3 RNAi disrupts the expression of PHP, while PHP persists in pre > Cul3 RNAi. **b** Quantification of mEPSC and quantal content values in the indicated genotypes after PhTx application normalized to baseline values (–PhTx: pre > Cul3 RNAi, *n* = 8; post > Cul3 RNAi, *n* = 13; + PhTx: pre > Cul3 RNAi, *n* = 8; post > Cul3 RNAi, *n* = 18). **c** Representative traces from the indicated genotypes and conditions showing a trans-heterozygous genetic interaction between *inc* and *Cul3* in PHP expression. While PHP is robustly expressed in *inc*[kk3]/ + or *Cul3*[EY11031]/ + alone, PHP is completely blocked in the trans-heterozygous condition. **d** Quantification of mEPSC and quantal content values after PhTx application normalized to baseline values (–PhTx: *inc*[kk3]/ + , *n* = 11; *Cul3*[EY11031]/ + , *n* = 8; *inc*[kk3]/ + ; *Cul3*[EY11031]/ + , *n* = 11; + PhTx: *inc*[kk3]/ + , *n* = 11; *Cul3*[EY11031]/ + , *n* = 8; *inc*[kk3]/ + ; *Cul3*[EY11031]/ + , *n* = 14; Student's *t*-test). **e** Representative NMJ images of wild-type and *inc* mutants immunostained with anti-pCaMKII (active phosphorylated CaMKII), -DLG (Discs Large; postsynaptic density marker) and -BRP (Bruchpilot; presynaptic active zone marker) before and after PhTx application. A similar reduction in pCaMKII levels are observed following PhTx application in both wild-type and *inc* mutants. In contrast, BRP levels are increased after PhTx application in wild type, but do not change after PhTx application to *inc* mutants, consistent with a lack of retrograde PHP signaling and expression. **f** Quantification of pCaMKII mean fluorescence intensity and BRP puncta sum intensity after PhTx application relative to wild-type values in the indicated genotypes (pCaMKII: –PhTx: wild type, *n* = 27; *inc*[kk3], *n* = 16; + PhTx: wild type, *n* = 14; *inc*[kk3], *n* = 19; BRP: –PhTx: wild type, *n* = 10; *inc*[kk3], *n* = 15; + PhTx: wild type, *n* = 11; *inc*[kk3], *n* = 10; one-way ANOVA). **g** Schematic illustrating postsynaptic Inc and Cul3 signaling in the induction of retrograde PHP expression. Asterisks indicate statistical significance: (**) *p* < 0.01; (***) *p* < 0.001; (****) *p* < 0.0001, (ns) not significant. Error bars indicate ± SEM. *n* values indicate biologically independent cells. Additional statistical information and absolute values for normalized data can be found in Supplementary Table 2. Source data are provided as a Source Data file

---

However, pCaMKII levels were similar at the NMJs of *inc* mutants and wild-type controls in baseline conditions, and were also reduced to similar levels following PhTx application (Fig. 3e, f and Supplemental Table 2).

Finally, retrograde signaling from the postsynaptic compartment leads to enhancement and remodeling of the presynaptic active zone scaffold Bruchpilot[10,17,22,34,35] (BRP). We considered that if retrograde signaling from the postsynaptic compartment was indeed disrupted in *inc* mutants, BRP should fail to be remodeled following PhTx application. BRP puncta intensity rapidly increased at presynaptic terminals following PhTx application at wild-type NMJs, as expected (Fig. 3e, f). However, no change in BRP puncta levels was observed in *inc* mutants following PhTx application (Fig. 3e, f). Altogether, these results demonstrate that *inc* functions downstream of or in parallel to CaMKII activity in the postsynaptic compartment, where it is necessary for emission of the retrograde signal that homeostatically modulates presynaptic structure and neurotransmitter release (schematized in Fig. 3g).

**Inc and Cul3 rapidly accumulate in postsynaptic compartments.** A localized reduction in active CaMKII is observed specifically at the postsynaptic density following genetic loss or pharmacological perturbation of glutamate receptors[17,20] (Fig. 3), suggesting that the key processes driving synapse-specific retrograde PHP signaling occur in this structure[23]. We therefore determined whether Inc is present at the postsynaptic density. We endogenously tagged *inc* with an smFP tag (*inc*[smFP]; see Methods) and verified that this tag does not disrupt basal synaptic transmission or PHP expression (Supplementary Fig. 3). Imaging of Inc[smFP] in the larval preparation revealed a low and diffuse cytosolic signal with some enrichment at the NMJ (Supplementary Fig. 4). Strikingly, we found that Inc[smFP] signal intensity increased rapidly at NMJs after perturbation of glutamate receptors using 10 min application of PhTx (Fig. 4a, c and Supplementary Fig. 4a). In contrast, the cytoplasmic Inc[smFP] signal in the muscle did not significantly change after PhTx (Supplementary Table 2).

Inc[smFP] is present at presynaptic motor neuron terminals in addition to the postsynaptic muscle and postsynaptic density (PSD). We therefore performed two experiments to assess whether the change in Inc[smFP] levels after PhTx incubation occurs in the postsynaptic compartment. First, we co-stained Inc[smFP] with a neuronal membrane marker (HRP) and a postsynaptic density marker (DLG). We considered the HRP

signal to label all of the neuronal compartment, and a fraction of the postsynaptic compartment, labeled by DLG, where HRP and DLG overlap in the embedded structure characteristic of the fly NMJ (Fig. 4b). A substantial fraction of the total Inc[smFP] signal (~75%) overlaps with both HRP and DLG, while the remaining proportion of the Inc[smFP] NMJ signal overlapped with DLG but not HRP (Fig. 4b, c and Supplementary Table 2). We found that the Inc[smFP] signal that overlapped with both HRP and DLG, and with DLG only, increased after PhTx application (Fig. 4a, c). These findings suggest that postsynaptic Inc[smFP] rapidly accumulates at NMJs after acute glutamate receptor perturbation. Second, we expressed an RNAi transgene targeting *inc*[24] in motor neurons or muscle in *inc*[smFP] backgrounds. NMJ levels of Inc[smFP] were reduced by 52% with presynaptic knock down, and reduced by 56% with postsynaptic knock down (Supplementary Table 2). Consistent with postsynaptic Inc[smFP] being enhanced at the PSD, a clear increase in the Inc[smFP] signal was observed at the NMJ after PhTx in preparations expressing presynaptic *inc* RNAi (Fig. 4d, e). Finally, we examined whether Cul3 behaved similarly after PhTx by expressing a UAS-3xFlag-3xHA-Cul3 transgene[36] exclusively in the postsynaptic muscle. At baseline, Flag-HA-Cul3 appeared diffuse across the muscle (Fig. 4f and Supplementary Fig. 4b). However, after 10 min PhTx application, Flag-HA-Cul3 rapidly accumulated around DLG at the NMJ (Fig. 4f, g and Supplemental Fig. 4b). Altogether, these experiments indicate that Cul3 and its adaptor Inc rapidly accumulate together at the postsynaptic density following diminished glutamate receptor functionality.

**Inc is required for local mono-ubiquitination during PHP.** The Cul3-Inc complex might target substrates for poly-ubiquitination and drive their degradation by the proteasome. Alternatively, Cul3-Inc may regulate substrates by non-degradative mechanisms, including mono-ubiquitination (schematized in Fig. 5b), a post translational modification that can modulate protein and membrane trafficking, as well as signaling[37,38]. A recent study rigorously explored the role of proteasomal degradation during PHP at the *Drosophila* NMJ[39]. Postsynaptic PHP signaling was not impacted by acute pharmacological or chronic genetic inhibition of proteasome-mediated protein degradation[39]. We therefore sought to determine whether the rapid accumulation of Cul3-Inc at the PSD is associated with mono- or poly-ubiquination of substrates.

We immunostained wild-type and *inc* NMJs with two anti-Ubiquitin antibodies at basal conditions and following 10 min

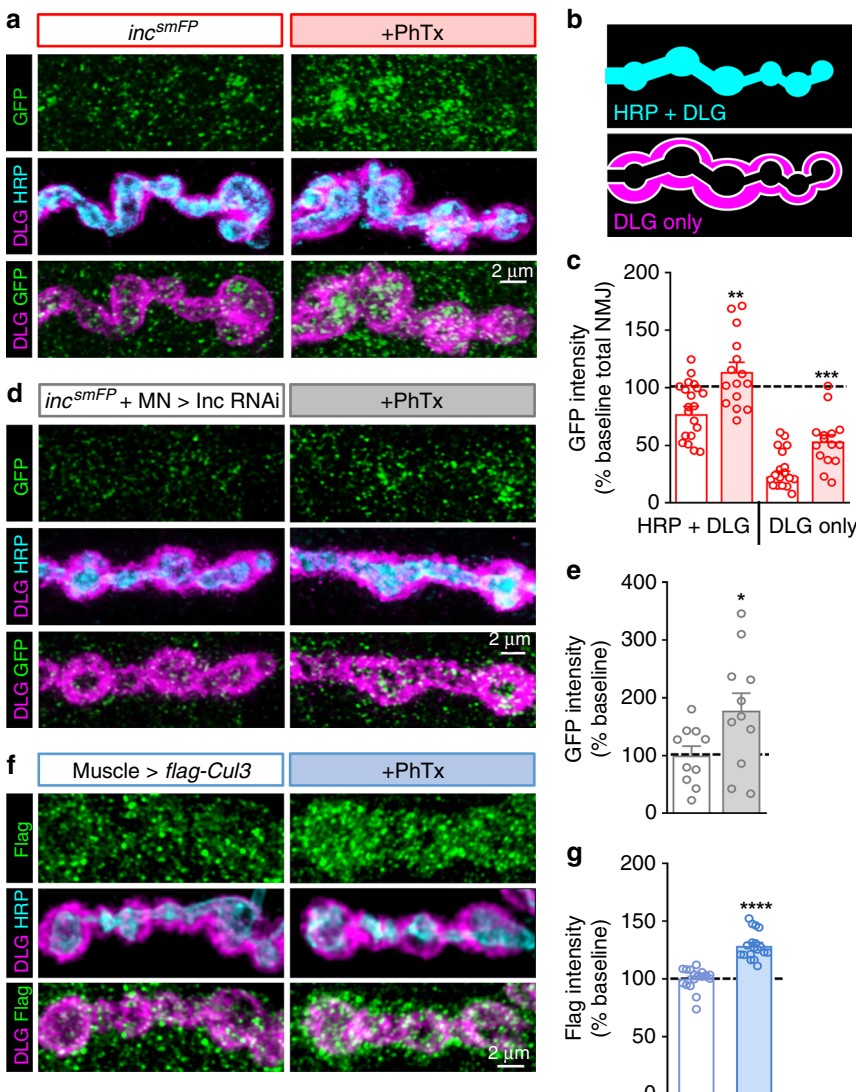

**Fig. 4** Inc and Cul3 levels are rapidly enhanced at postsynaptic densities following glutamate receptor perturbation. **a** Representative NMJ image of endogenously tagged Inc$^{smFP}$ before and after PhTx application. NMJs are immunostained with anti-GFP (Inc$^{smFP}$), the presynaptic membrane marker HRP, and the postsynaptic density marker DLG. Note the increase in Inc$^{smFP}$ levels at NMJs (labeled by HRP and DLG). **b** Schematic illustrating the HRP + DLG signal (where Inc$^{smFP}$ overlaps in both pre- and postsynaptic areas) and the DLG only component (where only postsynaptic Inc$^{smFP}$ overlaps with DLG). **c** Quantification of Inc$^{smFP}$ intensity at specified areas following PhTx application relative to baseline (–PhTx) (–PhTx: $inc^{smFP}$, $n = 21$; + PhTx: $inc^{smFP}$, $n = 15$). **d** NMJ image of endogenously tagged Inc$^{smFP}$ with $inc$ expression knocked down in the presynaptic motor neuron using Inc-RNAi ($inc^{smFP}/Y;OK6$-$Gal4/UAS$-$inc$ $RNAi$) before and after PhTx application. Note that while the Inc$^{smFP}$ signal that overlaps with HRP is reduced by 52% (Supplementary Table 2), a significant increase in the total level of Inc$^{smFP}$ at the NMJ is observed. **e** Quantification of Inc$^{smFP}$ intensity at NMJs in presynaptic $inc$ knock down following PhTx application relative to baseline (–PhTx) (–PhTx: $inc^{smFP}$ + MN > Inc RNAi, $n = 10$; + PhTx: $inc^{smFP}$ + MN > Inc RNAi, $n = 11$). **f** NMJ image of a Flag-tagged Cul3 transgene expressed in the postsynaptic muscle ($G14$-$Gal4/UAS$-$3xFlag$-$3xHA$-$Cul3$) before and after PhTx application. Note that Cul3 signals are enhanced at NMJs following 10 min PhTx application. **g** Quantification of Flag (Cul3) intensity at PSDs (labeled by DLG) following PhTx application relative to baseline (–PhTx) (–PhTx: muscle > $flag$-$Cul3$, $n = 19$; + PhTx: muscle > $flag$-$Cul3$, $n = 18$). Asterisks indicate statistical significance using a Student's $t$-test: (**) $p < 0.01$; (***) $p < 0.001$; (****) $p < 0.0001$, (ns) not significant. Error bars indicate ± SEM. $n$ values indicate biologically independent cells. Additional statistical information and absolute values for normalized data can be found in Supplementary Table 2. Source data are provided as a Source Data file

PhTx incubation. The FK2 antibody recognizes both poly- and mono-ubiquitinated proteins[40], while the FK1 antibody recognizes only poly-ubiquitinated conjugates[40]. We found that the ubiquitin signal labeled by FK2 rapidly increased at NMJs, with a large fraction of the FK2 signal localizing outside of HRP at postsynaptic densities following PhTx application (Fig. 5a, c). In contrast, no change in the FK1 signal was observed (Fig. 5a, d). This suggests that acute glutamate receptor perturbation increases

mono-ubiquitination at postsynaptic densities. However, no change in the FK2 or FK1 signal was observed at NMJs in $inc$ mutants following PhTx application (Fig. 5a, c, d), indicating that $inc$ is required for the rapid and compartmentalized increase in mono-ubiquitination at the NMJ following acute postsynaptic glutamate receptor perturbation. Importantly, postsynaptic, but not presynaptic, rescue of $inc$ mutants restored the increase in the FK2 signal after PhTx application (Fig. 5e, f). Thus, Inc and Cul3

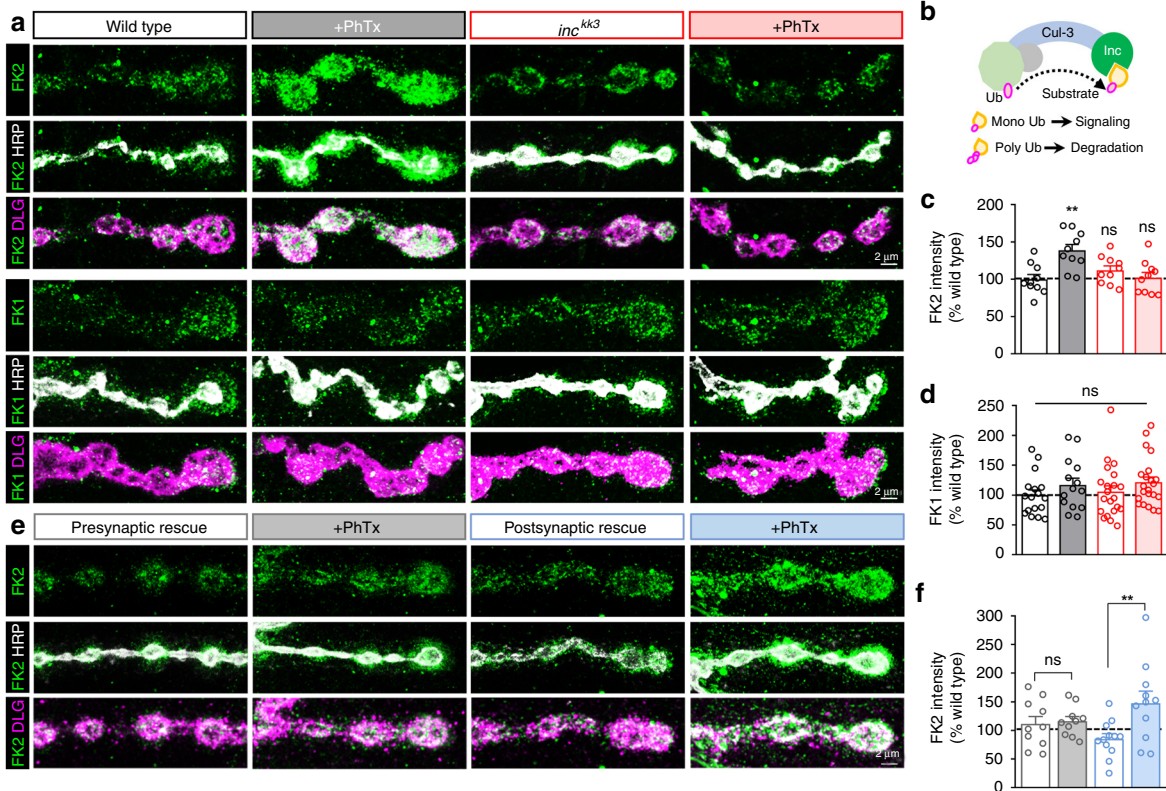

**Fig. 5** *inc* is required for rapid mono-ubiquitination at postsynaptic compartments following glutamate receptor perturbation. **a** Representative NMJ images from wild-type and *inc* mutants immunostained with anti-FK2 (mono- and poly-ubiquitin) or anti-FK1 (poly-ubiquitin only), -HRP and -DLG, before and after PhTx application. At wild-type NMJs, the FK2 signal rapidly increases at postsynaptic densities after PhTx application (indicated by the signal outside of HRP), while no change is observed in the FK1 signal. However, no change in either FK2 or FK1 intensity is observed at baseline or after PhTx application in *inc* mutant NMJs. **b** Schematic illustrating the Cul3 ubiquitin ligase complex, where Inc serves as an adaptor to target substrates for either mono-ubiquitination (which can confer signaling functions) or poly-ubiquitination (which leads to proteasomal degradation). Quantification of average FK2 (**c**) and FK1 (**d**) immunointensity levels after PhTx application in the indicated genotypes normalized to wild type (–PhTx: wild type, $n = 10$ (FK2), $n = 17$ (FK1); $inc^{kk3}$, $n = 10$ (FK2), $n = 21$ (FK1); + PhTx: wild type, $n = 10$ (FK2), $n = 14$ (FK1); $inc^{kk3}$, $n = 10$ (FK2), $n = 22$ (FK1)). **e** Representative NMJ images immunostained with anti-FK2, -HRP, and -DLG in *inc* mutants rescued presynaptically or postsynaptically at baseline and after PhTx application. While the FK2 signal fails to increase after PhTx in the presynaptic *inc* rescue condition, the FK2 signal increase is restored in the postsynaptic rescue. **f** Quantification of mean FK2 intensity in the indicated genotypes normalized to wild-type values (–PhTx: presynaptic rescue, $n = 10$; postsynaptic rescue, $n = 12$; + PhTx: presynaptic rescue, $n = 10$; postsynaptic rescue, $n = 11$). Asterisks indicate statistical significance using a one-way ANOVA followed by Tukey's multiple comparison test: (**) $p < 0.01$; (ns) not significant. Error bars indicate ± SEM. *n* values indicate biologically independent cells. Additional statistical information and absolute values for normalized data can be found in Supplementary Table 2. Source data are provided as a Source Data file

rapidly traffic to postsynaptic compartments and are required for the rapid accumulation of mono-ubiquintated proteins within minutes of glutamate receptor blockade during retrograde homeostatic signaling.

**Peflin links Ca²⁺-regulation, Cul3, and retrograde signaling.** Finally, we sought to gain insight into how Cul3/Inc-mediated mono-ubiquitination at postsynaptic compartments functions in the context of glutamate receptor perturbation and retrograde signaling. We assembled a list of 23 genes that have been shown to interact with Cul3, Inc, or with their mammalian orthologs (Supplementary Table 1). From this list, we obtained 17 mutants and ten RNAi lines targeting these genes to screen for rapid PHP expression with a strategy similar to that described in Fig. 1 (Fig. 6a and Supplementary Table 1). To identify genes functioning with Cul3 or Inc specifically in the postsynaptic compartment, we expressed the RNAi in muscle and screened both mutants and RNAi lines by assessing synaptic physiology after 10 min incubation in PhTx (Fig. 6a, b). This screen identified one mutant (a transposon insertion) and one RNAi line, both

targeting the same gene (annotated as *CG17765*), that failed to exhibit rapid PHP expression (Fig. 6a, b). Specifically, PhTx application to *CG17765* mutants or to larvae expressing *CG17765* RNAi postsynaptically reduced mEPSCs but did not increase quantal content, leading to reduced EPSC amplitude compared to baseline values and a block in PHP expression (Fig. 6d, e). Thus, loss of *CG17765* in the postsynaptic muscle is sufficient to disrupt retrograde PHP expression.

*CG17765* encodes the sole *Drosophila* ortholog of Peflin, a Ca²⁺-binding protein containing five repetitive penta-EF-hand motifs[41]. Mammalian Peflin was recently demonstrated to function as a co-adaptor for Cul3 with its BTB-domain containing adaptor KLHL12[38]. Pef heterodimerizes with another penta-EF-hand protein, ALG2, to impose Ca²⁺ regulation on Cul3-KLHL12. During neural crest specification, Ca²⁺ release from the ER in chondrocytes triggers Cul3/KLHL12/Pef/ALG2-dependent mono-ubiquitination of Sec31 to modulate Collagen secretion (schematized in Fig. 6c). Intriguingly, a homolog of Collagen XV/XVIII, *Drosophila* Multiplexin (Dmp), is necessary for retrograde PHP expression[14].

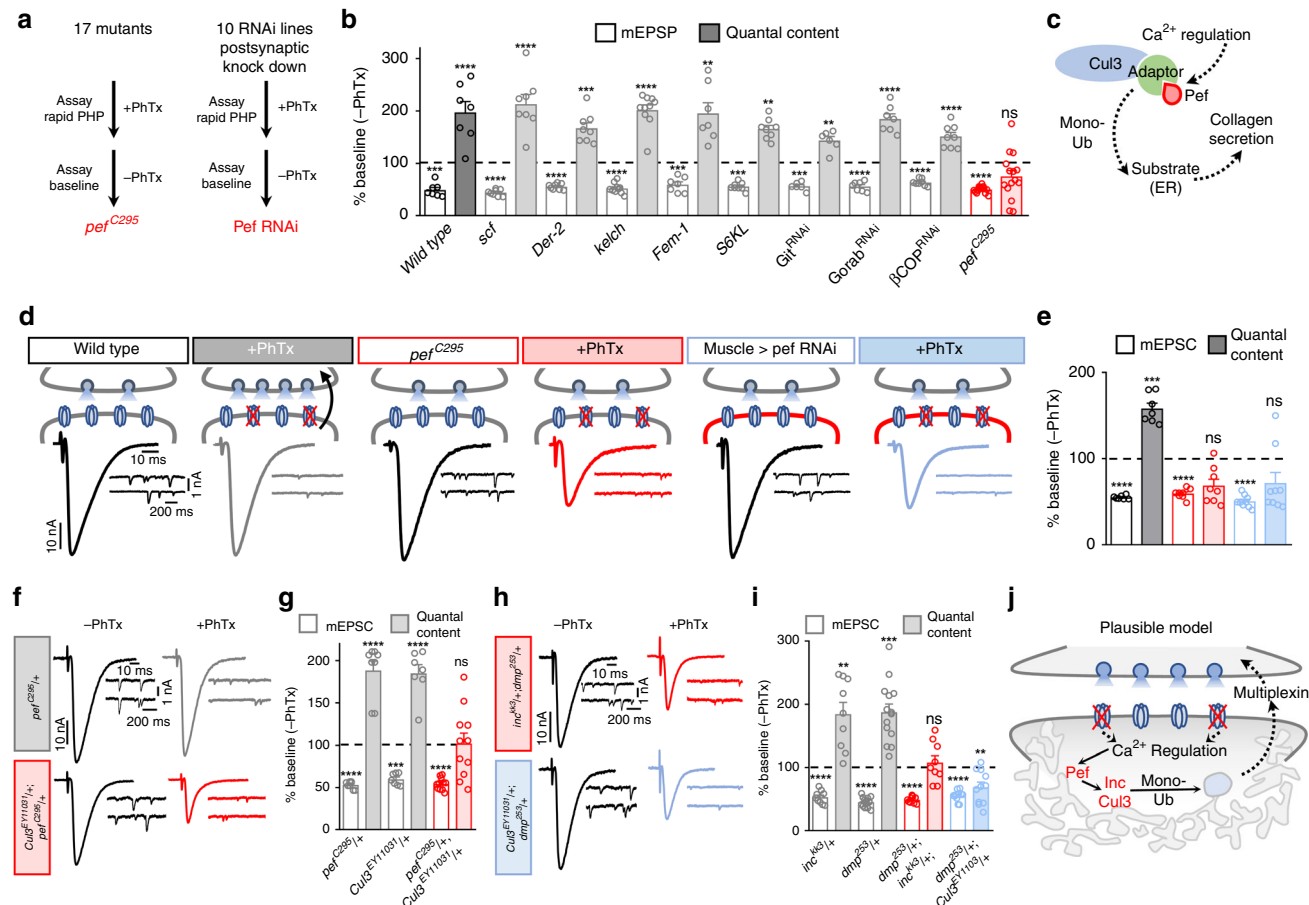

**Fig. 6** An Inc- and Cul3-interaction screen identifies *peflin* to be required postsynaptically for PHP expression. **a** Flow diagram and screening strategy of Inc- and Cul3-interacting genes identifies a mutation and RNAi line targeting *Drosophila peflin* (*CG17765; pef*). **b** Quantification of mEPSP and quantal content values in a subset of mutants screened normalized to baseline values (–PhTx) ($n = 6$–20; see Supplementary Table 1). **c** Schematic summarizing the role of mammalian Pef functioning as a co-adaptor for the Cul3-KLHL12 complex, which is regulated by $Ca^{2+}$ signaling. This signaling activates $Cul3^{KLHL12}$ to mono-ubiquitinate substrates involved in membrane trafficking at the ER to modulate Collagen secretion. **d** Schematic and representative traces of wild type, *peflin* mutants (*pef^{C295}*), and Pef RNAi driven in the postsynaptic muscle (muscle > Pef RNAi: *G14-Gal4/UAS-Pef RNAi*) before and following PhTx application. Note that while mEPSC amplitude is reduced in all three genotypes after PhTx application, PHP fails to be expressed in *pef^{C295}* and muscle > Pef RNAi. **e** Quantification of mEPSC and quantal content values in the indicated genotypes normalized to baseline values (–PhTx: wild type, $n = 8$; *pef^{C295}*, $n = 7$; muscle > Pef RNAi, $n = 10$; + PhTx: wild type, $n = 7$; *pef^{C295}*, $n = 8$; muscle > Pef RNAi, $n = 9$). Representative traces (**f** and **h**) and quantification (**g** and **i**) of genetic interaction experiments between *Cul3^{EY11031}/pef^{C295}* (*Cul3^{EY11031}/ +*, $n = 8$ ( - PhTx, + PhTx); *pef^{C295}/ +*, $n = 8$; *Cul3^{EY11031}/ +, pef^{C295}/ +*, $n = 21$), *inc^{kk3}/dmp^{253}* (*inc^{kk3}/ +*, $n = 8$; *dmp^{253}/ +*, $n = 8$; *inc^{kk3}/ +; dmp^{253}/ +*, $n = 9$) and *Cul3^{EY11031}/dmp^{253}* (*Cul3^{EY11031}/ +; dmp^{253}/ +*, $n = 10$). **j** Schematic illustrating a speculative but plausible model linking Cul3/Inc and Pef in retrograde homeostatic signaling. Asterisks indicate statistical significance using a Student's *t*-test: (**) $p < 0.01$; (***) $p < 0.001$; (****) $p < 0.0001$, (ns) not significant. Error bars indicate ± SEM. $n$ values indicate biologically independent cells. Additional statistical information and absolute values for normalized data can be found in Supplementary Table 2. Source data are provided as a Source Data file

To determine whether *Cul3/inc* and *pef* interact genetically in the context of PHP signaling, and whether they also interact with *dmp*, we assessed PHP expression in trans-heterozygous combinations of these mutations. This approach has been used previously to identify a relationship between several genes associated with PHP[14,15,18,42]. We found that PHP is robustly expressed following PhTx application in *inc^{kk3}/ +*, *Cul3^{EY11031}/ +*, and *pef^{C295}/ +* heterozygotes and confirmed that PHP is also expressed in *dmp^{253}/ +* larvae[14] (Fig. 6f–i). In contrast, PHP is blocked in *Cul3^{EY11031}/ +; pef^{C295}/ +* transheterozygous double mutants (Fig. 6f, g), consistent with *Cul3* and *pef* functioning in the same genetic pathway. Finally, we found that PHP failed to be expressed in *inc^{kk3}/ +; dmp^{253}/ +* and *Cul3^{EY11031}/ +; dmp^{253}/ +* transheterozygous double mutants (Fig. 6h, i). These data are consistent with *pef* and *Cul3/inc* functioning in the postsynaptic compartment,

perhaps to respond to changes in $Ca^{2+}$ signaling during homeostatic challenge, and to function with *dmp* in the same genetic pathway. A speculative but attractive possibility is that $Ca^{2+}$-dependent regulation of Pef may recruit and activate Cul3/Inc-dependent mono-ubiquitination to initiate the induction of local retrograde homeostatic signaling at postsynaptic compartments (schematized in Fig. 6j).

## Discussion

By screening >300 genes with putative functions at synapses, we have identified *inc* as a key postsynaptic regulator of retrograde homeostatic signaling at the *Drosophila* NMJ. Our data suggest that Inc and Cul3 are recruited to the postsynaptic compartment within minutes of glutamate receptor perturbation, where they

promote local mono-ubiquitination. Inc/Cul3 appear to function downstream of or in parallel to CaMKII and upstream of retrograde signaling during PHP. We identify Pef as a putative co-adaptor that may work with Inc/Cul3 to link $Ca^{2+}$ signaling in the postsynaptic compartment with membrane trafficking and retrograde communication. Altogether, our findings implicate a post translational signaling system involving mono-ubiquitination in the induction of retrograde homeostatic signaling at postsynaptic compartments.

Although forward genetic screens have been very successful in identifying genes required in the presynaptic neuron for the expression of PHP, these screens have provided less insight into the postsynaptic mechanisms that induce retrograde homeostatic signaling. It seems clear that many genes acting presynaptically are individually required for PHP[8,9,12,30,42], with loss of any one completely blocking PHP expression. Indeed, ~25 genes that function in neurons have thus far been implicated in PHP expression[3,7]. In contrast, forward genetic screens have largely failed to uncover new genes functioning in the postsynaptic muscle during PHP, implying some level of redundancy. The specific postsynaptic induction mechanisms driving retrograde PHP signaling have therefore remained unclear[16,17], and are further complicated by cap-dependent translation and metabolic pathways that contribute to sustaining PHP expression over chronic, but not acute, time scales[17,29,43]. Therefore, it is perhaps not surprising that despite screening hundreds of mutants, we found only a single gene, *insomniac*, to be required for PHP induction. Inc is expressed in the nervous system and can traffic to the presynaptic terminals of motor neurons[32]. In the context of PHP signaling, however, we found *inc* to be required in the postsynaptic compartment, where it functions downstream of or in parallel to CaMKII. One attractive possibility is that a reduction in CaMKII-dependent phosphorylation of postsynaptic targets enables subsequent ubiquitination by Cul3-Inc complexes, and that this modification ultimately drives retrograde signaling during PHP. Indeed, reciprocal influences of phosphorylation and ubiquitination on shared targets are a common regulatory feature in a variety of signaling systems[44]. The dynamic interplay of phosphorylation and ubiquitination in the postsynaptic compartment may enable a sensitive and tunable mechanism for controlling the timing and calibrating the amplitude of retrograde signaling at the NMJ.

The substrates targeted by Inc and Cul3 during PHP induction are not known, but the identification of mono-ubiquitination in the postsynaptic compartment during PHP signaling and the putative Cul3 co-adaptor Peflin provides a foundation from which to assess possible candidates and pathways. In mammals, Pef forms a complex with another $Ca^{2+}$ binding protein, ALG2, to confer $Ca^{2+}$ regulation to membrane trafficking pathways[45,46]. Moreover, Pef/ALG2 were recently found to serve as target-specific co-adaptors for Cul3-KLHL12[38]. In particular, SEC31 and other components involved in ER-mediated membrane trafficking pathways were shown to be targeted for mono-ubiquitination, which in turn modulate Collagen secretion[38,47]. One attractive possibility, therefore, is that Cul3/Inc could respond to changes in $Ca^{2+}$ in the postsynaptic compartment through regulation by Pef during PHP signaling to control membrane trafficking pathways. Importantly, the subsynaptic reticulum (SSR) is a complex and membraneous network at the *Drosophila* NMJ, where electrical, $Ca^{2+}$-dependent, and membrane trafficking pathways in the postsynaptic compartment are integrated[48,49]. Indeed, Multiplexin, a fly homolog of Collagen XV/XVIII and a proposed retrograde signal, is secreted into the synaptic cleft and is required for trans-synaptic retrograde signaling during PHP[14]. In addition, another proposed retrograde signal and secreted protein, Semaphorin 2B, was recently shown to function post-synaptically in retrograde PHP signaling[15]. However, *inc* does not

appear to be the closest *Drosophila* ortholog to KLHL12, and it is therefore possible that Pef and Cul3/Inc regulate postsynaptic PHP signaling through a more indirect mechanism.

While the precise relationships between CaMKII, Inc, Cul3, and Pef are currently unclear, the activity of membrane trafficking pathways could ultimately be targeted for modulation by $Ca^{2+}$- and Cul3/Inc-dependent signaling during PHP induction. First, a role for postsynaptic membrane trafficking and elaboration during PHP signaling has already been suggested[18,50]. In addition, extracellular $Ca^{2+}$ does not appear to be involved in rapid PhTx-dependent PHP induction[17]. It is therefore tempting to speculate that $Ca^{2+}$ release from the postsynaptic SSR during rapid PHP signaling may influence Cul3/Inc activity through Pef-dependent regulation, as transient changes in ER-derived $Ca^{2+}$-signaling controls Pef-dependent recruitment of Cul3[38]. Alternatively, postsynaptic scaffolds and/or glutamate receptors themselves may be targeted by Cul3/Inc at the *Drosophila* NMJ, given that these proteins are involved in ubiquitin-mediated signaling and remodeling at dendritic spines[37,51]. Consistent with this idea, there is evidence that signaling complexes composed of neurotransmitter receptors, CaMKII, and membrane-associated guanylate kinases are intimately associated at postsynaptic densities in *Drosophila*[52], as they are in the mammalian central nervous system[53]. There has been speculation that these complexes are targets for modulation during PHP signaling[17,20]. Although these models are not mutually exclusive, further studies will be required to determine the specific substrates and signal transduction mechanisms through which Cul3/Inc and Pef initiate and sustain retrograde homeostatic communication in postsynaptic compartments.

While it is well established that the ubiquitin proteasome system can sculpt and remodel synaptic architecture, the importance of mono-ubiquitination at synapses is less studied. Ubiquitin-dependent pathways play key roles in synaptic structure, function, and degeneration, and also contribute to activity-dependent dendritic growth[54–56]. However, the fact that some proteins persist for long periods at synapses suggests that modification of these proteins by ubiquitin likely include non-degradative and reversible mechanisms. Indeed, a recent study revealed a remarkable heterogeneity in the stability of synaptic proteins, with some short lived and rapidly turned over, while others persisting for long time scales, with half lives of months or longer[57]. At the *Drosophila* NMJ, rapid ubiquitin-dependent proteasomal degradation at presynaptic terminals is necessary for the expression of PHP through modulation of the synaptic vesicle pool[39]. In contrast, postsynaptic proteasomal degradation does not appear to be involved in rapid PHP signaling, suggesting that ubiquitin-dependent pathways in the postsynaptic compartment contribute to PHP signaling by non-degradative mechanisms. Our data demonstrate that Cul3, Inc, and Pef function in muscle to enable retrograde PHP signaling, and suggest that Cul3/Inc rapidly trigger mono-ubiquitination at postsynaptic densities following glutamate receptor perturbation. Interestingly, synaptic proteins can be ubiquitinated in <15 s following depolarization-induced $Ca^{2+}$ influx at synapses[58] and changes in intracellular $Ca^{2+}$ can activate Pef and Cul3 signaling with similar rapidity[38]. Therefore, both poly- and mono-ubiquination may function in combination with other rapid and reversible processes, including phosphorylation[18,19] at postsynaptic compartments to enable robust and diverse signaling outcomes during the induction of homeostatic plasticity.

A prominent hypothesis postulates that a major function of sleep is to homeostatically regulate synaptic strength following experience-dependent changes that accrue during wakefulness[59]. Several studies have revealed changes in neuronal firing rates and synapses during sleep/wake behavior[60–65], yet few molecular mechanisms that directly associate the electrophysiological

process of homeostatic synaptic plasticity and sleep have been identified. Our finding that *inc* is required for the homeostatic control of synaptic strength provides an intriguing link to earlier studies, which implicate *inc* in the regulation of sleep[24,25]. It remains to be determined to what extent the role of *inc* in controlling PHP signaling at the NMJ is related to the impact of *inc* on sleep and, if so, whether Inc targets the same substrates to regulate these processes. Interestingly, virtually all neuropsychiatric disorders are associated with sleep dysfunction, including those associated with homeostatic plasticity and Fragile X Syndrome[66], and sleep behavior is also disrupted by mutations in the *Drosophila* homolog of FMRP, *dfmr1*[67]. Further investigation of this intriguing network of genes involved in the homeostatic control of sleep and synaptic plasticity may help solve the biological mystery that is sleep and also shed light on the etiology of neuropsychiatric diseases.

## Methods

**PHP screen.** We identified over 800 mammalian genes that encode transcripts expressed at synapses and that have not been previously screened for PHP. This list was generated from recent studies that identified putative transcripts associated with FMRP (see Supplementary Data 1). This list was further supplemented with an additional 176 genes associated with schizophrenia and autism spectrum disorder. From these initial lists of mammalian genes, we identified 352 *Drosophila* homologs using NCBI protein Basic Local Alignment Search Tool (BLAST) algorithm with an *E*-value cutoff of $10^{-3}$. We used a combination of previously characterized genetic mutations and/or transposon mutations (197) or RNA-interference transgenes (341) targeting these genes to obtain a stock collection to screen. Finally, we assessed the lethal phase of homozygous mutants and RNAi lines crossed to motor neuron and muscle Gal4 drivers, and also determined if transmission phenotypes matched expectations for the subset of genes with known roles in synaptic function. We eliminated any mutants or RNAi lines that failed to survive to at least the third-instar larval stage or show the expected phenotype. This led to a final list of 124 mutations to screen blinded by PhTx application and 249 RNAi lines to screen blinded by GluRIII knock down (Supplementary Data 1). The RNAi screen was performed using T15 and C15 lines[31].

**Fly stocks.** All *Drosophila* stocks were raised at 25 °C on standard molasses food and obtained from the Bloomington Drosophila Stock Center unless otherwise noted. The following fly stocks were used in this study: T15 and C15[31]; *inc-Gal4*, *inc[1]* and *inc[2]* [24]; *OK371-Gal4*; *MHC-Gal4*; *UAS-Cul3 RNAi[11861R-2]* (Fly Stocks of National Institute of Genetics); *UAS-Dcr2*; *GluRIIA[SP16]*; *G14-Gal4*; *BG57-Gal4*; *UAS-3xHA-3xFlag-Cul3[36]*; *OK6-Gal4*; *dmp[f07253]* [14]; *UAS-Pef RNAi[v32404]* (Vienna Drosophila Resource Center; VDRC); *pef[C295]* (CG17765[C295]), *Cul3[EY11031]*, *Df(1)Exel8196*, *UAS-inc-RNAi[v18225]* (VDRC), and *UAS-CD4-td-eGFP*. The *w[1118]* strain was used as the wild-type control unless otherwise noted because this is the genetic background in which all genotypes are bred. See Supplementary Table 1 and Supplementary Data 1 for sources of the screened mutants and RNAi lines.

**Molecular biology.** *inc[kk]* mutants were generated using a CRISPR/Cas-9 genome editing strategy[68,69]. Briefly, we selected a target Cas-9 cleavage site in the first coding exon of *inc* without obvious off-target sequences in the *Drosophila* genome (sgRNA target sequence: 5′ GTTCCTCTCCCGTCTGATTC <u>AGG</u> 3′, PAM underscored). DNA sequences containing this target sequence were synthesized and subcloned into the pU6-BbsI-chiRNA plasmid (45946; Addgene, Cambridge, MA). To generate the sgRNA, pU6-BbsI-chiRNA was PCR amplified and cloned into the pattB vector. This construct was injected and inserted into the attP40 target sequence on the second chromosome and balanced. This line was crossed into a stock expressing Cas-9 under control of *vas* regulatory sequences, which led to 9 independent indels with predicted frameshift mutations in the *inc* open reading frame confirmed by PCR followed by sequencing of the *inc* locus in male flies after balancing. Lines that introduced the earliest stop codon (R50Stop) and the second earliest stop codon (C57STOP) were chosen for further analyses and were named *inc[kk3]* and *inc[kk4]* respectively. These mutants were then outcrossed for eight generations to ensure new isogenic second and third chromosomes were incorporated, to control for the genetic background, and to eliminate potential off-target effects of Cas-9 activity.

To generate *UAS-smFP-inc*, we subcloned the full-length *inc* cDNA from the expressed sequence tag LD43051 (Drosophila Genomics Resources Center; Bloomington, IN) into the pACU2 vector (31223; Addgene, Cambridge, MA) using standard methods. A spaghetti monster FLAG tag[33] (10xFLAGsmFP) was PCR amplified and placed in-frame before the stop codon of the *inc* open reading frame. Constructs were sequence verified and injected into the *w[1118]* strain using the VK18 insertion site on the second chromosome by BestGene Inc. (Chino Hill, CA). Endogenously tagged *inc[smFP]* was generated by Well Genetics Inc. (Taipei, Taiwan) using CRISPR/Cas-9 targeting and homology directed repair. Briefly, a construct

containing the smFP-10xFLAG as well as a 3xP3 DsRed reporter was inserted just before the stop codon of the endogenous *inc* locus using a single target gRNA synthesized as RNA. This construct was injected into a *w[1118]* strain with Cas-9 expression and the insertion was confirmed by DsRed+ eyes; the DsRed marker was subsequently excised using the pBac transposase, leaving only smFP at the *inc* C-terminus. The insertion was confirmed by genomic PCR sequencing.

**Electrophysiology.** For all two-electrode voltage clamp (TEVC) recordings, muscles were clamped at −70 mV, with a leak current below 5 nA[12,70]. mEPSCs were recorded for 1 min from each muscle cell in the absence of stimulation. Twenty EPSCs were acquired for each cell under stimulation at 0.5 Hz, using 0.5 ms stimulus duration and with stimulus intensity adjusted with an ISO-Flex Stimulus Isolator (A.M.P.I.). To acutely block postsynaptic receptors, larvae were incubated with or without philanthotoxin-433 (PhTx; 20 μM; Sigma) in HL-3 for 10 min[8,21]. Data were analyzed using Clampfit 10.7 (Molecular Divices), MiniAnalysis (Synaptosoft), Excel (Microsoft), and GraphPad Prism (GraphPad Software).

**Immunocytochemistry.** Third-instar larvae were dissected in 0 Ca$^{2+}$ HL-3 and immunostained[71]. All genotypes were immunostained in the same tube with identical reagents, and mounted in the same session. The following antibodies were used: mouse anti-Bruchpilot (BRP; nc82; 1:100; Developmental Studies Hybridoma Bank; DSHB); mouse anti-GluRIIA (1:100; 8B4D2; DSHB); guinea pig anti-GluRIID[72] (1:1000); rabbit anti-DLG[17] (1:5000); mouse anti-DLG (1:100; 4F3; DSHB); mouse anti-FK1 (1:100; Millipore 04–262); mouse anti-FK2 (1:500; BML-PW8810; Enzo Life Sciences); mouse anti-FLAG (1:500, F1804; Sigma-Aldrich); mouse anti-GFP (1:1000, 3e6; Invitrogen, Carlsbad, CA); mouse anti-pCaMKII (1:100; MA1–047; Invitrogen). Donkey anti-mouse, anti-guinea pig, and anti-rabbit Alexa Fluor 488- (715-545-150, 706-545-148, 711-545-152; Jackson Immunoresearch), DyLight 405- (715-475-150, 706-475-148, 711-475-152; Jackson Immunoresearch), and Cyanine 3 (Cy3)- (715-165-150, 706-165-148, 711-165-152; Jackson Immunoresearch) conjugated secondary antibodies were used at 1:400. Alexa Fluor 647 conjugated goat anti-HRP (123-605-021; Jackson ImmunoResearch) was used at 1:200.

**Western blot.** Protein extracts were prepared from male whole animals by homogenization in ice-cold NP40 lysis buffer (50 mM Tris pH7.6, 150 mM NaCl, 0.5% NP40) supplemented with protease inhibitors (Sigma, P8340). Protein lysates were centrifuged at 4 °C at 15,000 x *g* for 15 min and quantitated in duplicate (BioRad, 5000111). Sixty micrograms of protein were resolved by Tris-sodium dodecyl sulfate polyacrylamide gel electrophoresis and transferred to nitrocellulose. Membranes were blocked for 1 h at room temperature in LI-COR Odyssey buffer (LI-COR, 927–40000). Membranes were subsequently incubated overnight at 4 °C in blocking buffer containing 0.1% Tween 20, rat anti-Insomniac[24] (1:1000), and mouse anti-tubulin (1:100,000, Genetex, gtx628802). After washing 4 × 5 min in TBST (150 mM NaCl, 10 mM Tris pH7.6, and 0.1% Tween 20), membranes were incubated in the dark for 30 min at room temperature in blocking buffer containing 0.1% Tween 20, 0.01% SDS, Alexa 680 donkey anti-rat (1:30,000, Jackson ImmunoResearch, 712-625-153), and Alexa 790 donkey anti-mouse (1:30,000, Life Technologies, A11371). Membranes were washed 4 × 5 min in TBST, 1 × 5 min in TBS, and imaged on a LI-COR Odyssey CLx instrument.

**Sleep behavior.** One- to 4-day-old flies eclosing from cultures entrained in LD cycles (12 h light/12 h dark) were loaded into glass tubes and assayed for 5–7 days at 25 °C in LD cycles using DAM2 monitors (Trikinetics). Male flies were assayed on food containing cornmeal, agar, and molasses. Female flies were assayed on food containing 5% sucrose and 2% agar. The first 36–48 h of data were discarded, to permit acclimation and recovery from CO$_2$ anesthesia, and an integral number of days of data (3–5) were analyzed using custom Matlab software[24]. Locomotor data was collected in 1 min bins, and a 5 min period of inactivity was used to define sleep; a given minute was assigned as sleep if the animal was inactive for that minute and the preceding 4 min. Dead animals were excluded from analysis by a combination of automated filtering and visual inspection of locomotor traces.

**Confocal imaging and analysis.** Briefly, samples were imaged using a Nikon A1R Resonant Scanning Confocal microscope equipped with NIS Elements software using a 100x APO 1.4NA or 60 × 1.4NA oil immersion objective. All genotypes were imaged in the same session with identical gain and offset settings for each channel across genotypes. z-stacks were obtained using identical settings for all genotypes, with z-axis spacing between 0.15 and 0.5 μm within an experiment and optimized for detection without saturation of the signal[73]. Maximum intensity projections were used for quantitative fluorescence intensity analysis with the NIS Elements software General Analysis toolkit. We compared summation of signal intensities from individual images to the maximum intensity projection and did not find significant differences in the ultimate result. All quantifications were performed for Type Ib boutons on muscle 6/7 and muscle 4 of segments A2 and A3. Type Ib boutons were selected at individual NMJs based on DLG intensity. Measurements were taken from at least ten synapses acquired from at least six different animals. For all images, fluorescence intensities were quantified by applying intensity thresholds to eliminate background signal. For analysis of BRP

fluorescence levels, the sum intensity of individual BRP puncta was quantified. For analysis of pCaMKII and Inc$^{smFP}$ intensity levels, a mask was created around the DLG and HRP channels, and a DLG-HRP mask was used to define the exclusively postsynaptic region; only Inc$^{smFP}$ signals within this mask were quantified. For FK1 and FK2 anti-Ubiquitin staining, mean fluorescence intensity was calculated using regions with the DLG-HRP mask (to exclusively assess the postsynaptic area).

**Statistical analysis**. Data were compared using either a one-way analysis of variance (ANOVA) and followed by Tukey's multiple comparison test, or using a Student's *t*-test (where specified), analyzed using Graphpad Prism or Microsoft Excel software, and with varying levels of significance assessed as $p < 0.05$ (*), $p < 0.01$ (**), $p < 0.001$ (***), $p < 0.0001$ (****), ns = not significant. For statistical analysis of sleep duration, one-way ANOVA and Tukey-Kramer post hoc tests were used. For all figures, data are quantified as averages + /− SEM, and absolute values and additional statistical details are presented in Supplementary Table 2.

**Reporting summary**. Further information on research design is available in the Nature Research Reporting Summary linked to this article.

## Data availability
The data that support the findings of this study are available from D.D. upon reasonable request. The authors declare that the data supporting the findings of this study are available within the paper, the Data Source file, and the Supplementary Information and Supplementary Data files.

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

## Acknowledgements

We thank C. Andrew Frank (University of Iowa, USA), Mark Tanouye (University of California, Berkeley, USA), Lynn Cooley (Yale University, USA), and Gabrielle Boulianne (University of Toronto, Canada) for sharing fly stocks, and Martin Müller (University of Zürich, Switzerland) for insightful discussions and comments. We acknowledge the Developmental Studies Hybridoma Bank (Iowa, USA) for antibodies used in this study and the Bloomington Drosophila Stock Center for fly stocks (NIH P40OD018537). K.K. was supported in part by a USC Provost Fellowship. Q.L. was supported by an International Student Research Fellowship from the Howard Hughes Medical Institute (HHMI). This work was supported by a grant from the Mathers Foundation, Whitehall Foundation grant 2013-05-78, fellowships from the Alfred P. Sloan and Leon Levy Foundations, a NARSAD Young Investigator Award from the Brain and Behavior Foundation, the J. Christian Gillin, M.D. Research Award from the Sleep Research Society Foundation, a Career Scientist Award from the Irma T. Hirschl/Weill-Caulier Trust and by a grant from the National Institutes of Health (NS111304) to N.S.; and by a grant from the National Institutes of Health (NS091546) and fellowships from the Mallinckrodt, Whitehall, and Klingenstein-Simons Foundations to D.D.

## Author contributions

K.K. and D.D. conceived the project and designed the research. K.K., X.L., S.P., P.G., C.C., D.K., and Q.L. performed experiments. K.K., X.L., P.G., C.C., S.P., and Q.L. analyzed data. K.K. and D.D. wrote the manuscript with feedback from X.L., S.P., P.G., Q.L., and N.S.

## Additional information

**Competing interests:** The authors declare no competing interests.

