## [Peer Review File · Nature Communications]

Reviewers' comments:

Reviewer #1 (Remarks to the Author):

The manuscript by Dickman and colleagues describes a role for a mono-ubiquitination pathway mediated by the insomniac protein for postsynaptic regulation of a form of homeostatic plasticity at *Drosophila* neuromuscular synapses. The authors identify insomniac from a set of screens for mutants in homeostatic plasticity. They then generate CRISPR-mediated mutants of the locus and show a role for the protein in the postsynaptic muscle for the induction of homeostatic plasticity and the subsequent upregulation of the presynaptic active zone protein BRP. The mutant does not alter the previously described downregulation of postsynaptic CAMKII during this form of plasticity, suggesting it acts at a different step. The authors describe an increase in insomniac protein at the NMJ during plasticity, and provide data the protein acts together with the Cul-3 complex to drive mono-ubiquitination of postsynaptic targets as a means to control homeostatic plasticity. The intriguing link of the insomniac protein to sleep dysfunction in flies suggests some provocative possibilities for tying together these two processes, though this link is not explored. Overall the data is interesting and reveals another player (insomniac) and process (monoUb) in the postsynaptic cell required for this form of plasticity. How this process works to alter the transmission of a signal is unknown. I like the work on the insomniac mutant and find the increased localization at synapses to be compelling – though it is unclear how much of this is presynaptic or postsynaptic. The screen itself has some problems, and I wish the authors knew more about the postsynaptic targets of mono-ubiquitylation that could reveal new insights there.

Other comments:

1. A key component of the authors model is that the biology they are describing is postsynaptic, given the prior work implicating proteasome function presynaptically. Several areas of the paper are not clear in this regard. The pre-/post-rescue data indicates a postsynaptic role. However, the link to the 2nd half of the paper is less clear. Fig. 4A shows increased insomniac staining at the NMJ with an endogenously tagged version, but this increase looks presynaptic – the staining is contained within the anti-DLG staining and appears to be inside the presynaptic bouton. The authors could use pre-/post-synaptic RNAi (since they have an RNAi line that works from their screen, or they could use an anti-GFP RNAi, as this will knockdown smFP as well) to prove this increased accumulation is indeed postsynaptic. If this is presynaptic accumulation as the image shows to me, then this would be consistent with some function in the presynaptic terminal as well not related to homeostatic plasticity, and potentially alter the model for what is occurring postsynaptically. The authors need to use pre-/post-specific knockdown to show this accumulation is indeed postsynaptic, where the authors are arguing the biology is occurring.

2. The localization of monoUb proteins is also unclear in Fig. 4D, where the FK2 label appears to increase again mostly in the presynaptic compartment. This can be easily shown to be postsynaptic by the RNAi knockdown of insomniac in the pre- and post-synaptic RNAi experiments done in Fig. 2. Does the FK2 staining go away with muscle knockdown as it should, or is this instead an unrelated presynaptic effect as the image appears to show. The localization can't really be determined by the immunostaining, but can easily be distinguished by compartment specific knockdown of ins, with FK2 staining after PhTx addition. This is a key missing experiment.

3. It would be very nice to do a mono-Ub IP (with the FK2 antibody), or mass-spec, or westerns with the FK2 antibody to try and identify some of the potential PSD targets of this pathway. As it stands, we're left with an interesting possibility, but no real targets for transmitting the retrograde signal. This would be a really nice boost to the story. Another approach might be to look for changes in endocytic markers like Rab5 in the PSD, given the link between monoUb and receptor internalization.

4. Does the tagged insomniac protein re-localize to the PSD in CAMKII strains that block

homeostatic plasticity? That would help the authors put the protein in a clearer pathway.

5. Does Cul-3 become enriched in the postsynaptic compartment as well, as predicted? There are labeled Cul-3 lines that could be used for this analysis.

6. Why did the authors not include the FK1 staining experiment in the inc mutants in panel 4d – this should be included to assay this change in inc mutants before and after PhTx.

7. I think the authors need to clean up their description of the screen that starts things off. Many of the RNAi knockdown lines they tested simply did not work in eliminating the gene of interest, and therefore can't be assumed to have been screened. Looking at the included table, this sticks out all over the place – a few striking examples: Cacophony (quantal content (QC) of 60 – should have no release), (LAP – mini amplitude of 0.4 mv – reported phenotype of much bigger minis), para (QC of 70, should be embryonic lethal with no action potentials, so no release), syntaxin (QC of 75, should have no release). Many others in the table are also simply not giving the phenotype reported in the field from more careful mutant analysis that has been done. As such, the RNAi is not working for many genes. Any fly synaptic person who reads that list is going to see many of their favorite genes that they are interested in looking for their involvement in homeostatic plasticity and be disappointed, as they will see no effect. However, when they look at the baseline physiology the authors report here, they will see it doesn't match to phenotypes in the field, and therefore will have real skepticism about the screen results the authors present. I think the authors need to include a column on their table that would indicate whether the RNAi result is even trustable, or to remove all those lines and indicate they don't match up with the published physiology in the field. For the genes that have not been studied in the field, they obviously can't do that, but it raises concern about how many of those RNAis can be trusted. Overall, I feel the authors can use the screen as a discovery approach – negative hits have no meaning they can give to the data (unless they verify the gene knockdown), while positive hits can be tested further, as they did for insomniac. They need to indicate this issue right up front so everyone knows what is going on. To give the assumption they have really tested the function of all these genes listed in PHP is incorrect, as many of the RNAis are simply not working (I don't know if this is 10%, 50% or 90%), and the authors provide no data (Q-PCR, western, etc.) to validate any of these putative knockdowns. As such, Figure 1 is largely useless as its unclear which of those hits on the x-axis is real or not. I recommend beginning the story with Figure 2, and leaving the table (with some modifications) to just indicate what lines they tested so there is no implication for the negative data (unless they do validation to show the knockdown is actually working). It's fine to use RNAi screens as a means to identify an interesting gene to follow up on --- it's not fine to imply that the RNAi screen rules out the vast majority of genes where the RNAi is simply not working. These can be easily fixed in the re-write.

8. The discussion of the screen isn't just a problem with the RNAi part -- for example, I'm surprised that the AZ Fife came out of the authors' screen given their criteria that baseline synaptic transmission is not affected. The authors cite a Fife mutant paper (Bruckner et al, 2017) - that study showed baseline synaptic transmission is reduced by greater than 50% for both evoked amplitude (Fig. 1G) and quantal content (Fig. 1H). How could this pass the criteria the authors put forward for their "hits" – loss of PHP and no change in baseline synaptic transmission. Clearly, Fife mutants fail to qualify as a hit based on their criteria – why did the authors include it? Is the Bruchner paper wrong on their claim that Fife mutants have such dramatic baseline effects on synaptic transmission? Several other mutants also jump off that list that don't match published physiology data from the field. Again, this is going to have a negative effect on people looking at those gene lists, if the few examples cited don't fit with the authors' stated criteria.

Reviewer #2 (Remarks to the Author):

Using a forward genetic and electrophysiological screen, Kikuma et al report the identification of Inc as a key postsynaptic regulator of retrograde signaling for presynaptic homeostatic potentiation (PHP) at the *Drosophila* NMJ. Their data suggest that following pharmacological or genetic reduction of postsynaptic glutamate receptor function, Inc is rapidly recruited to the postsynaptic density, where it may function as an adaptor for the Cullin-3 (Cul3) E3 ubiquitin ligase complex, promoting synaptically localized ubiquitination. The project design is very smart. Importantly, most of the experiments were well executed, and represent important, albeit incremental, insights into core mechanisms required for PHP. However, the work remains very much premature and there are loopholes and hand-waving. A major revision with a significant amount of additional experimentations will be required for increasing the merits of the present work for being considered for publication in the journal.

Major concerns:

1. The rapid recruitment of Inc into the synapse following the acute blockade of glutamate receptors is interesting, but how does the receptor blockade induce such a rapid translocation? Is the Inc itself mono-ubiquitinated, and translocated into the synapses thereafter? What are other major substrate(s) for Inc-Cul3 E3-ligase complex?
2. The major challenge for understanding the core mechanisms of PHP is to explain how reduced glutamate receptor activity at the postsynaptic site is translated into the increase in presynaptic release. While the present work identifies Inc as a necessary component of the postsynaptic mechanism, it adds little on how the retrograde trans-synaptic signaling is accomplished. Revealing how Inc-Cul3 is involved in generating the retrograde signaling would significantly increase the impact of the work.

Minor points:

1. Fig. 2g: There should be a WT control group.
2. Fig. 3d: are the values of the reduction in pCamKII between PhTx-treated and non-treated indeed similar, but not statistically different? A proper statistic comparison has to be performed before the authors can claim they are similar.
3. Fig. 4a: Is the increase of postsynaptic Inc following PhTx treatment a result of Inc translocation from the soma to synapses or a global increase?

Reviewer #3 (Remarks to the Author):

NCOMMS-18-30182

The sleep gene *insomniac* ubiquitinates targets at postsynaptic densities and is required for retrograde homeostatic signaling

In this manuscript, Kikuma et al. investigate a new role for the gene *insomniac* in retrograde synaptic signaling. The authors start by performing a forward genetic screen and identify *insomniac* as a modulator of retrograde signaling (after the silencing of the GluRII receptors). Besides its relevance for sleep maintenance, *insomniac* is essential in the fly muscle to activate the retrograde signaling from the muscle to the motor neuron, acting either downstream or in parallel to CaMKII. Finally, the authors focused on the molecular function of *insomniac* as an E3 ubiquitin ligase and showed that *insomniac* is relevant for the mono-ubiquitination of synaptic proteins. This is an interesting paper that presents a new player in neuronal plasticity. There are however a couple of problems that I hope the authors can address. These relate to the difficulty in interpretation of the results that are not straightforward due to lacking data, the improper reporting of statistical analyses, and an imprecise description of the methodology.

1. Around 400 genes were chosen to start the genetic screen. However, since several diseases present a defect in the homeostatic control of the synaptic plasticity, I am a bit unsure as to the justification of the choice of FMPR targets to create this collection. Can this be clarified?
2. The authors show that the loss of insomniac results in a lack of ubiquitination of proteins at the synapse. However, there is no direct link to the altered retrograde signaling observed in the mutants. In line with the previous comments, a broader screen (that contains a larger number of genes) could help to elucidate the downstream events that link insomnia loss to the function of the presynaptic neuron.
3. The authors opted for a graphical representation of the results of the screen instead of showing the individual data for each of the genes that can modulate the retrograde signaling (without changing the basal synaptic strength). While this option allows a rapid visualization of the results, it is important to present the identity of these genes (in supplementary material for example). In addition, it is important to present the data used for the classification of the positive genes in pre- or post-synapse (lines 142-144).
4. While the new insomniac mutants (kk3 and kk4) seem to have the same phenotypes as the available insomniac mutants (1 and 2), the creation of the mutants needs to be better controlled. Were the mutants checked for off targets of the CRISPR/Cas9 activity? Did the authors check the insomniac mRNA levels?
5. Can the authors comment on the apparent contradiction between the supplementary figure 1 (where the EPSC, mEPSC and quantal content are altered) and the statement on the maintenance of the basal synaptic transmission of inc mutants (lines 119-123 and 158-159)?
6. The methods and figure legends need to be improved. For example:
 - a. Fig 1S g and h: Are all the animals used females? If so, please confirm that the genotypes of the orange bar is "inc1/inc1, inc-Gal4" and the red is "inc1/inc1, inc-Gal4; UAS-smFP-inc/+". Also, can the authors justify the difference of the total daily sleep in the wild-type flies in g and h? A value of 1000 minutes of sleep is typical for male flies and not for females.
 - b. Can the authors clarify the method to quantify the intensity of the fluorescent signal in figure 3? According to the methods section, the signal of a z-stack was projected using the maximum intensity function, which is a non-standard method to quantify those images. Instead, a sum of all the imaged z-planes is used to quantify the fluorescent signal.
 - c. Please provide the method used to match the mammalian and Drosophila homologs.
 - d. Which type of Cas-9 was used? Which method was used to test for off-targets (line 363)?
 - e. Please provide a detailed description of the statistical test to each graph, since it is difficult to understand some of the comparisons. For example, figure 4c, is the ns compared to the 100%? Also, is the *** a comparison to the 100% or to the value of the black column? In other words, was it used a one-sample t-test in this case?
7. Please provide improved images for the post-synaptic location of the endogenous inc-smFP, since the image provided in figure 4a does not clearly show the presence of insomniac in that structure.
8. What is the link between the lack of mono-ubiquitination at the post-synapse (in the insomnia mutants after PhTx) and the decreased retrograde signaling? Also, the authors failed to demonstrate that the levels of ubiquitin are not altered at the synapse (which could be a cause for the decreased protein ubiquitination in the mutants) or to define the proteins that are targets of insomniac at the post-synapse that play a role in this mechanism.

RESPONSE TO REVIEWERS

We thank the three Reviewers for their time and efforts in providing constructive comments for our manuscript. We are encouraged by the overall positive assessments from all three Reviewers and appreciate the valid and helpful suggestions raised to improve the clarity and impact of our study. The central comments revolved around **1)** technical concerns about the pre- vs post-synaptic compartment in which Inc levels are enhanced and functions in mono-ubiquitination, and **2)** more insight into how Inc enables retrograde homeostatic signaling. We have focused on addressing these and other outstanding issues in this thoroughly revised manuscript.

First, we have performed the key controls suggested by the Reviewers to demonstrate that postsynaptic Inc increases at the NMJ after homeostatic induction. We also show that *inc* expression is necessary in the postsynaptic muscle, but not presynaptic neuron, to enhance the Ubiquitin signal at the PSD following PhTx application. Further, we now show that Cul3, exclusively expressed in the muscle, is also enhanced at PSD after PhTx, providing an independent line of evidence that levels of Inc and its partner Cul3 become rapidly increased at the PSD after homeostatic signaling. These additional experiments and controls have greatly strengthened and clarified the pre- vs. post-synaptic localizations and functions of Inc and Cul3.

Second, we have worked over the past five months to provide more insight into how Inc/Cul3-mediated signaling drives retrograde homeostatic communication in the postsynaptic compartment. We are very excited by the new insights this effort has achieved, which we expect to establish a new foundation from which to understand the induction of presynaptic homeostatic potentiation and to have major impacts in the field. As described in more detail below, we have undertaken a targeted screen of Inc- and Cul3-interacting genes for postsynaptic roles in PHP. This effort has identified the Ca²⁺-regulated Cul3 co-adaptor Peflin to be necessary in the postsynaptic muscle to drive PHP signaling. A recent study published in *Cell* reported that Peflin imposes Ca²⁺ regulation on Cul3 and its adaptor to localize to ER structures and mono-ubiquitinate a substrate, ultimately modulating membrane trafficking of collagen and its secretion (McGourty et al., 2016). Intriguingly, we now show that *Cul3*, *inc*, and *peflin* genetically interact with each other and with a candidate retrograde signal, *multiplexin* (Wang et al., 2014), a member of the collagen family in the context of PHP expression. Although significant additional work is clearly necessary to elucidate how Cul3, Inc, Peflin, and mono-ubiquitination function in the postsynaptic compartment to enable retrograde homeostatic signaling, these new results begin to provide a framework from which to understand how altered Ca²⁺ signaling in the postsynaptic density, triggered by diminished GluR function, may control the secretion of Multiplexin and other retrograde signaling systems during PHP.

Finally, we have provided extensive additional controls, analyses, and textual revisions to improve the clarity as suggested by the Reviewers. These results and new data are included in additions to Figures 3, 4 and 5 and an entirely new Figure 6, along with new Supplemental data.

Together, these efforts have greatly improved the manuscript, and we hope the Reviewers now deem this revision to be appropriate for publication in *Nature Communications*.

RESPONSE TO REVIEWER 1

The manuscript by Dickman and colleagues describes a role for a mono-ubiquitination pathway mediated by the insomniac protein for postsynaptic regulation of a form of homeostatic plasticity at Drosophila neuromuscular synapses. The authors identify insomniac from a set of screens for mutants in homeostatic plasticity. They then generate CRISPR-mediated mutants of the locus and show a role for the protein in the postsynaptic muscle for the induction of homeostatic plasticity and the subsequent upregulation of the presynaptic active zone protein BRP. The mutant does not alter the previously described downregulation of postsynaptic CAMKII during this form of plasticity, suggesting it acts at a different step. The authors describe an increase in insomniac protein at the NMJ during plasticity, and provide data the protein acts together with the Cul-3 complex to drive mono-ubiquitination of postsynaptic targets as a means to control homeostatic plasticity. The intriguing link of the insomniac protein to sleep dysfunction in flies suggests some provocative possibilities for tying together these two processes, though this link is not explored. Overall the data is interesting and reveals another player (insomniac) and process (monoUb) in the postsynaptic cell required for this form of plasticity. How this process works to alter the transmission of a signal is unknown. I like the work on the insomniac mutant and find the increased localization at synapses to be compelling – though it is unclear how much of this is presynaptic or postsynaptic. The screen itself has some problems, and I wish the authors knew more about the postsynaptic targets of mono-ubiquitylation that could reveal new insights there.

1. A key component of the authors model is that the biology they are describing is postsynaptic, given the prior work implicating proteasome function presynaptically. Several areas of the paper are not clear in this regard. The pre-/post-rescue data indicates a postsynaptic role. However, the link to the 2nd half of the paper is less clear. Fig. 4A shows increased insomniac staining at the NMJ with an endogenously tagged version, but this increase looks presynaptic – the staining is contained within the anti-DLG staining and appears to be inside the presynaptic bouton. The authors could use pre-/post-synaptic RNAi (since they have an RNAi line that works from their screen, or they could use an anti-GFP RNAi, as this will knockdown smFP as well) to prove this increased accumulation is indeed postsynaptic. If this is presynaptic accumulation as the image shows to me, then this would be consistent with some function in the presynaptic terminal as well not related to homeostatic plasticity, and potentially alter the model for what is occurring postsynaptically. The authors need to use pre-/post-specific knockdown to show this accumulation is indeed postsynaptic, where the authors are arguing the biology is occurring.

We thank the Reviewer for making this point and we agree that given the difficulty is parsing pre- vs. post-synaptic Inc^{smFP} localization, the additional experiments suggested by the Reviewer would help demarcate which compartment Inc^{smFP} accumulates. Clearly, Inc^{smFP} is present at both presynaptic motor neuron terminals and throughout the muscle, including postsynaptic densities (Fig. 4 and S4). To delineate in which compartment the increased Inc^{smFP} is observed after PhTx application, we have performed three new experiments that indicate that it is the postsynaptic fraction of Inc^{smFP} and Cul3 that become enriched in the postsynaptic density. First, we have carefully analyzed the overlap of Inc^{smFP} with both the HRP and DLG signal (HRP+DLG, which consists of both pre- and post-synaptic Inc^{smFP}) vs the Inc^{smFP} that overlaps with only the DLG signal. This quantifies the fraction of Inc^{smFP} that is exclusively postsynaptic (diagrammed in revised Fig. 4 and discussed in **lines 251-253**. We see a

significant enhancement of Inc^{smFP} after PhTx in the DLG only area (Fig. 4a, c), consistent with postsynaptic Inc^{smFP} becoming enriched at the PSD.

Second, we used RNAi-mediated knock-down of *inc* in presynaptic motor neurons to reduce the Inc^{smFP} signal specifically in the presynaptic compartment, as suggested by the Reviewer. While *inc* RNAi expressed pre- or post-synaptically did not reduce Inc expression enough to impact PHP expression (**lines 179-181** and Supplementary Table 3), this approach did reduce presynaptic Inc^{smFP} by 56%, enabling a clearer analysis of how postsynaptic Inc^{smFP} changes after PhTx. This approach also found that postsynaptic Inc^{smFP} is enhanced after PhTx application; results are now shown in Fig. 4d, e.

Finally, Inc serves as an adapter for the Cul3 ubiquitin ligase, and we tested whether Cul3 levels also change in the postsynaptic muscle after PhTx application. Importantly, we expressed a Flag-tagged Cul3 transgene exclusively in the muscle and examined levels at baseline and after PhTx application. Importantly, the entire signal of Cul3 in this experiment is postsynaptic, eliminating the challenges in assigning pre- vs post-synaptic localization. This experiment revealed a remarkable change in Cul3 after PhTx compared to baseline, with an obvious enhancement in the postsynaptic compartment. This data is now shown in Fig. 4f, g and Supplemental Fig. 4b.

These experiments provide compelling and independent lines of evidence for postsynaptic levels of Inc and Cul3 increasing at the PSD after PhTx, and we thank the Reviewer for these important suggestions.

2. The localization of monoUb proteins is also unclear in Fig. 4D, where the FK2 label appears to increase again mostly in the presynaptic compartment. This can be easily shown to be postsynaptic by the RNAi knockdown of insomniac in the pre- and post-synaptic RNAi experiments done in Fig. 2. Does the FK2 staining go away with muscle knockdown as it should, or is this instead an unrelated presynaptic effect as the image appears to show. The localization can't really be determined by the immunostaining, but can easily be distinguished by compartment specific knockdown of *inc*, with FK2 staining after PhTx addition. This is a key missing experiment.

We agree this is an important experiment that should be performed. As suggested by the Reviewer, we have now performed FK2 staining at the NMJ at baseline and after PhTx application in *inc* mutants rescued with pre- or post-synaptic *inc* expression. As expected, this revealed that the FK2 signal failed to increase after PhTx application in *inc* mutants rescued presynaptically, while the FK2 signal was enhanced in *inc* mutants rescued postsynaptically. These results are now shown in a revised Figure 5 and discussed in the Results (**lines 287-288**). We thank the Reviewer for suggesting this important experiment.

3. It would be very nice to do a mono-Ub IP (with the FK2 antibody), or mass-spec, or westerns with the FK2 antibody to try and identify some of the potential PSD targets of this pathway. As it stands, we're left with an interesting possibility, but no real targets for transmitting the retrograde signal. This would be a really nice boost to the story. Another approach might be to look for changes in endocytic markers like Rab5 in the PSD, given the link between monoUb and receptor internalization.

Defining the substrates of Inc and insight into how Inc/Cul3 ultimately drives retrograde homeostatic signaling, as pointed out by the other Reviewers, is of course of major interest. Although performing an anti-FK2 IP and performing mass spec may be an approach taken in

the future to identify potential Inc substrates, this strategy is not trivial, requires significant time to optimize and complete, and is likely to provide many candidates that are not specific to homeostatic plasticity and therefore require extensive validation. Given these limitations and caveats, we have elected to perform a candidate screen of potential Inc- and Cul3-interacting genes. It was this effort that necessitated the extension of two months to submit this revised manuscript, but we are very excited at the new insights this approach has brought to our manuscript.

Briefly, we assembled a list of 23 candidate genes that interact with *Drosophila* Insomniac or Cul3, or their mammalian homologs from a variety of proteomic and other studies (Bennett et al., 2010; Giot et al., 2003; Hein et al., 2015; Hudson et al., 2015; Hutchins et al., 2010; McGourty et al., 2016; Wang et al., 2011). We obtained genetic mutations and/or RNAi lines to screen for PHP expression as described in Fig. 1. From this effort, we identified the Ca²⁺-regulated Cul3 co-adaptor Peflin to be necessary in the postsynaptic muscle to drive PHP signaling. A recent study published in *Cell* reported that mammalian Peflin imposes Ca²⁺ regulation on Cul3 and its adaptor to localize to ER structures and mono-ubiquitinate a substrate to ultimately modulate the membrane trafficking pathways that control Collagen secretion (McGourty et al., 2016). Intriguingly, we now show that *Cul3*, *inc*, and *peflin* genetically interact with each other and with *multiplexin*, a member of the collagen family, in the context of PHP expression. Multiplexin is secreted into the synaptic cleft of the *Drosophila* NMJ, where it has been proposed to serve as a trans-synaptic signal mediating retrograde PHP signaling (Wang et al., 2014). Although significant additional work is clearly necessary to elucidate how Cul3, Inc, Peflin, and mono-ubiquitination function in the postsynaptic compartment to enable retrograde homeostatic signaling, these new insights begin to provide a framework from which to understand how altered Ca²⁺ signaling in the postsynaptic density, triggered by diminished GluR function, may control the secretion of Multiplexin and other retrograde signaling systems during PHP. These results are now presented in an entirely new Figure 6 and discussed in the revised Results and Discussion. We feel this effort has significantly expanded insights into the postsynaptic inductive pathway mediated by Inc during PHP signaling. We thank all three Reviewers for pushing us to gain more insight into the how Inc/Cul3 mediates retrograde homeostatic signaling.

Reviewer Figure 1: Endogenously tagged Rab5 does not significantly change at the NMJ after PhTx incubation. (a) Representative image of endogenously tagged GFP-Rab5 costained with HRP and DLG. **(b)** Schematic illustrating how the fraction of GFP-Rab5 that overlaps with both HRP and DLG was separated from the fraction that overlaps with DLG only. **(c)** Quantification of GFP-Rab5 intensity at specified areas following PhTx application relative to baseline (-PhTx). Error bars indicate \pm SEM.

We have also investigated whether the endocytic marker Rab5 changes at the NMJ following PhTx application, as suggested by the Reviewer. We obtained an endogenously tagged *rab5* allele that has been shown to retain native function and localization (Koles et al., 2015; Rodal et al., 2011). Rab5-GFP is present at the NMJ and appears to be enriched in the postsynaptic compartment (see Reviewer Fig. 1 above). We performed a similar analysis as shown in Fig. 4 of the Rab5 signal within both HRP and DLG (HRP+DLG) vs the faction that overlapped with DLG only. This revealed no change in the Rab5 signal after PhTx application compared to

baseline. However, we do not believe this experiment can definitively rule out a potentially important role for Rab5 in PHP signaling, so a future study will be necessary to clarify the membrane trafficking events that control homeostatic signaling in the postsynaptic density and subsynaptic reticulum. We have therefore decided not to include this data in the revised manuscript. We thank this Reviewer for suggesting this experiment.

4. Does the tagged insomniac protein re-localize to the PSD in CAMKII strains that block homeostatic plasticity? That would help the authors put the protein in a clearer pathway.

This is a very interesting idea. The Reviewer correctly points out that when constitutively active CaMKII is overexpressed in the postsynaptic muscle in *GluRIIA* mutants, PHP fails to be expressed (Haghighi et al., 2003; Li et al., 2018). It would indeed be of interest to determine whether Inc^{smFP} is still enhanced at the PSD in this condition after PHP induction. We have performed this experiment by applying PhTx to *inc^{smFP};G14-Gal4/UAS-CaMKII^{T287D}* flies. We find that Inc^{smFP} is no longer increased at the PSD after PhTx application (see Reviewer Figure 2). This suggests that a reduction in CaMKII activity may be necessary to recruit IncsmFP to the PSD. However, Flag-Cul3 appears to already be enriched at the PSD at baseline in CaMKII^{T287D} NMJs (~130% increased over Flag-Cul3 alone), and after PhTx appears to be reduced (~70% of -PhTx; Reviewer Fig. 2). Importantly, we have unpublished data that suggests that postsynaptic overexpression of CaMKII^{T287D} perturbs the PSD in significant but at present poorly understood ways. We therefore do not feel we can properly interpret the results of this experiment at the current time, and propose to not include this data in the manuscript so as to allow sufficient time to better understand this result. However, if the Reviewer insists, we can include this data as an additional Supplementary Figure.

Reviewer Figure 2: Inc^{smFP} and Flag-Cul3 localization in muscle>CaMKII^{T287D}. Representative NMJ image (a) and quantification (b) of endogenously tagged Inc^{smFP} before and after PhTx application in CaMKII^{T287D} muscle overexpression (*inc^{smFP};Y;G14-Gal4/UAS-CaMKII^{T287D}*). Representative NMJ image (c) and quantification (d) of Flag-Cul3 and CaMKII^{T287D} expressed in muscle (*G14-Gal4/UAS-CaMKII^{T287D};UAS-Flag-Cul3*) at specified areas following PhTx application relative to baseline (-PhTx).

5. Does Cul-3 become enriched in the postsynaptic compartment as well, as predicted? There are labeled Cul-3 lines that could be used for this analysis.

As the Reviewer indicates, Inc serves as an adaptor for the Cul3 ubiquitin ligase. Since Inc is enhanced at the PSD after PHP induction, it is a reasonable question to ask whether Cul3 is also increased. In addition, as the Reviewer points out, a Flag/HA-tagged Cul-3 transgene is available under the control of UAS elements (Hudson and Cooley, 2010). As suggested by the Reviewer, we performed this experiment and observed an obvious and robust enhancement in the Cul3 signal expressed exclusively in the postsynaptic muscle after PhTx. These results are described in more detail in Point 1 above and presented in the revised Fig. 4. This became a very strong line of evidence that it is postsynaptic Inc and Cul3 that are enhanced at the PSD

after PHP induction. This turned out to be an outstanding experiment and we greatly appreciate (and are indebted to) the astute suggestion by this Reviewer.

6. Why did the authors not include the FK1 staining experiment in the *inc* mutants in panel 4d – this should be included to assay this change in *inc* mutants before and after PhTx.

Because FK1 staining does not increase after PhTx application in controls, we did not include FK1 staining in *inc* mutants. However, as suggested by the Reviewer, we have now included this experiment in the revised Fig. 5. As expected, we observe no significant change in FK1 staining in *inc* mutants compared to wild type, either at baseline or following PhTx application.

7. I think the authors need to clean up their description of the screen that starts things off. Many of the RNAi knockdown lines they tested simply did not work in eliminating the gene of interest, and therefore can't be assumed to have been screened. Looking at the included table, this sticks out all over the place – a few striking examples: Cacophony (quantal content (QC) of 60 – should have no release), (LAP – mini amplitude of 0.4 mv – reported phenotype of much bigger minis), para (QC of 70, should be embryonic lethal with no action potentials, so no release), syntaxin (QC of 75, should have no release). Many others in the table are also simply not giving the phenotype reported in the field from more careful mutant analysis that has been done. As such, the RNAi is not working for many genes. Any fly synaptic person who reads that list is going to see many of their favorite genes that they are interested in looking for their involvement in homeostatic plasticity and be disappointed, as they will see no effect. However, when they look at the baseline physiology the authors report here, they will see it doesn't match to phenotypes in the field, and therefore will have real skepticism about the screen results the authors present. I think the authors need to include a column on their table that would indicate whether the RNAi result is even trustable, or to remove all those lines and indicate they don't match up with the published physiology in the field. For the genes that have not been studied in the field, they obviously can't do that, but it raises concern about how many of those RNAis can be trusted. Overall, I feel the authors can use the screen as a discovery approach – negative hits have no meaning they can give to the data (unless they verify the gene knockdown), while positive hits can be tested further, as they did for *insomniac*. They need to indicate this issue right up front so everyone knows what is going on. To give the assumption they have really tested the function of all these genes listed in PHP is incorrect, as many of the RNAis are simply not working (I don't know if this is 10%, 50% or 90%), and the authors provide no data (Q-PCR, western, etc.) to validate any of these putative knockdowns. As such, Figure 1 is largely useless as its unclear which of those hits on the x-axis is real or not. I recommend beginning the story with Figure 2, and leaving the table (with some modifications) to just indicate what lines they tested so there is no implication for the negative data (unless they do validation to show the knockdown is actually working). It's fine to use RNAi screens as a means to identify an interesting gene to follow up on --- it's not fine to imply that the RNAi screen rules out the vast majority of genes where the RNAi is simply not working. These can be easily fixed in the re-write.

We absolutely agree with the Reviewer on these points and have made several changes in the presentation and discussion of the screen in the revised manuscript. As the Reviewer points out, the screen we performed was used as a discovery approach, with candidate hits ultimately needing to be validated by generating new mutant alleles and characterized as we have done with *inc* in this manuscript. That being said, we want to discuss a few important points about the

screen and its limitations. As the Reviewer points out, for novel genes or genes that have never been characterized in the context of synaptic function, we cannot say one way or the other whether the mutants or RNAi lines were effective. We do, however, note that we have used the latest generation and most effective RNAi lines where available (TRiP 20), and have selected genetic mutations with transposon insertions in exonic regions, predicted to severely disrupt expression.

As suggested by the Reviewer, we have made two important revisions in the current manuscript regarding the screen. First, we have separated all genes screened into two categories **1)** Genes in which roles in synaptic function have not previously been assessed and/or do not impact basal synaptic function, and **2)** Genes in which mutations are known to disrupt baseline synaptic transmission. We include an additional column indicating whether the mutation or RNAi produced the expected phenotype in synaptic function. Importantly, we have removed all mutants and RNAi lines that are not consistent with the expected phenotype. This is now shown in the revised Supplemental Table 1. In addition, we provide a discussion in the revised Results stating these caveats and limitations to our screening approach, which is now detailed in **lines 124-128**. We explicitly state that for the majority of the mutants and RNAi lines screened that showed no defect in PHP, we cannot conclude these genes do not have a role in PHP. We thank the Reviewer for these suggestions which have helped to clarify our study.

8. *The discussion of the screen isn't just a problem with the RNAi part -- for example, I'm surprised that the AZ Fife came out of the authors' screen given their criteria that baseline synaptic transmission is not affected. The authors cite a Fife mutant paper (Bruckner et al, 2017) - that study showed baseline synaptic transmission is reduced by greater than 50% for both evoked amplitude (Fig. 1G) and quantal content (Fig. 1H). How could this pass the criteria the authors put forward for their "hits" – loss of PHP and no change in baseline synaptic transmission. Clearly, Fife mutants fail to qualify as a hit based on their criteria – why did the authors include it? Is the Bruchner paper wrong on their claim that Fife mutants have such dramatic baseline effects on synaptic transmission? Several other mutants also jump off that list that don't match published physiology data from the field. Again, this is going to have a negative effect on people looking at those gene lists, if the few examples cited don't fit with the authors' stated criteria.*

We agree on these points as well and thank the Reviewer for making these suggestions. For the vast majority of the mutants screened, either nothing was known about roles in basal transmission or it was previously demonstrated they do not reduce baseline transmission. This is clearly laid out in the revised Results (**lines 105-108**) and the revised Supplementary Table 1. Our data on *fife*, now clearly shown in Table 1, is consistent with published work on this mutant in PHP. Because of the design of the screen, it is not known which mutants are baseline transmission vs. PHP mutants until physiology with and without PhTx is assessed. Therefore, mutants which impact baseline transmission will necessarily be identified. However, we prioritized mutants that do not impact baseline transmission, which is why *fife* was not studied further. As the Reviewer notes, a few of the genes we screen do indeed have defects in baseline transmission, which are now explicitly noted in the Table, and we have revised the text and criteria for our screen accordingly (**lines 454-456**).

RESPONSE TO REVIEWER 2

Using a forward genetic and electrophysiological screen, Kikuma et al report the identification of Inc as a key postsynaptic regulator of retrograde signaling for presynaptic homeostatic potentiation (PHP) at the Drosophila NMJ. Their data suggest that following pharmacological or genetic reduction of postsynaptic glutamate receptor function, Inc is rapidly recruited to the postsynaptic density, where it may function as an adaptor for the Cullin-3 (Cul3) E3 ubiquitin ligase complex, promoting synaptically localized ubiquitination. The project design is very smart. Importantly, most of the experiments were well executed, and represent important, albeit incremental, insights into core mechanisms required for PHP. However, the work remains very much premature and there are loopholes and hand-waving. A major revision with a significant amount of additional experimentations will be required for increasing the merits of the present work for being considered for publication in the journal.

1. The rapid recruitment of Inc into the synapse following the acute blockade of glutamate receptors is interesting, but how does the receptor blockade induce such a rapid translocation? Is the Inc itself mono-ubiquitinated, and translocated into the synapses thereafter? What are other major substrate(s) for Inc-Cul3 E3-ligase complex?

25 μ g of total protein in INPUT; 250 μ g of total protein used for IP

Reviewer Figure 3: Inc^{smFP} is immunoprecipitated and blotted with anti-FK2 (bottom blots). No obvious increase in the FK2 signal is observed after PhTx application.

This is an intriguing idea and one we had not considered. First, we have quantified the Inc^{smFP} in muscle at NMJs and outside NMJs at baseline and following PhTx. As outlined in more detail in Minor Point 3 below, we do not observe a significant change in the Inc^{smFP} signal outside of NMJs after PhTx application (Supplemental Table 3 and **lines 242-243**). To specifically test whether Inc is mono-ubiquitinated after PhTx, we have immunoprecipitated Inc^{smFP} from larval muscle at baseline and after PhTx application and immunoblotted with anti-FK2. This experiment found no evidence that Inc^{smFP} itself is ubiquitinated after PhTx application, as the FK2 signal did not change (see Reviewer Figure 3).

In terms of other substrates targeted by Inc/Cul3, we agree this is a key question and one of major interest. To gain insight into this question, we have performed the genetic screen of Cul3- and Inc-interacting targets described in the response to Reviewer 1 point 3 above. The results of this approach have identified a co-adaptor, Peflin, and suggest putative substrates linked to secretory membrane trafficking processes may be targeted (including Sec31). Future studies will be focused on assessing and characterizing these potential substrates as well as the role of membrane trafficking at the postsynaptic compartment

during retrograde PHP signaling.

2. The major challenge for understanding the core mechanisms of PHP is to explain how reduced glutamate receptor activity at the postsynaptic site is translated into the increase in presynaptic release. While the present work identifies Inc as a necessary component of the postsynaptic mechanism, it adds little on how the retrograde trans-synaptic signaling is accomplished. Revealing how Inc-Cul3 is involved in generating the retrograde signaling would significantly increase the impact of the work.

We agree this is a question of major interest. First, given how little is known about the signal transduction mechanisms controlling PHP in the postsynaptic compartment, despite over 20 years of research into PHP at the fly NMJ, we strongly feel the identification of Inc and the process of mono-ubiquitination are major insights on their own. Second, it is difficult to establish a complete answer to this question within a reasonable timeframe for the revision of a manuscript. However, given these points, we have focused our efforts into uncovering as much insight as possible to understand how Inc and Cul3 are involved in generating retrograde homeostatic signaling.

We have performed a new screen of Inc-interacting genes now presented in a new Fig. 6 and detailed in Reviewer 1 Point 3 above. We have also performed additional experiments that have suggested links between Inc- and Cul3-related signaling and a putative retrograde signal necessary for trans-synaptic PHP expression recently proposed, Multiplexin (Wang et al., 2014). These experiments are detailed in the Response to Reviewer 1 Point 3 above and have improved the impact of our study. We thank the Reviewers for these suggestions.

Response to Minor Points by Reviewer 2

1. Fig. 2g: There should be a WT control group.

We have now added the WT control data to Fig. 2g. We apologize for this omission.

2. Fig. 3d: are the values of the reduction in pCamKII between PhTx-treated and non-treated indeed similar, but not statistically different? A proper statistic comparison has to be performed before the authors can claim they are similar.

We agree that the direct comparison of the reduction in the pCaMKII signal between PhTx-treated and non-treated in wild type vs *inc* mutants is important. We have performed an ANOVA statistical test and find no statistical difference between these data sets. This is now presented in Supplemental Table 3. We thank the Reviewer for pointing this out.

5. Fig. 4a: Is the increase of postsynaptic Inc following PhTx treatment a result of Inc translocation from the soma to synapses or a global increase?

We have examined the Inc^{smFP} signal at baseline and after PhTx at the NMJ as well as across the rest of the postsynaptic muscle cell. While we observe the increase in the Inc^{smFP} signal at NMJs, we do not observe a significant change at cytosolic regions of the muscle. Therefore, it does not appear to be a global increase in Inc^{smFP} expression, but rather an increase restricted to the NMJ. This data is now discussed in the Results (**lines 240-243**) and presented in Supplemental Table 3.

RESPONSE TO REVIEWER 3

In this manuscript, Kikuma et al. investigate a new role for the gene insomniac in retrograde synaptic signaling. The authors start by performing a forward genetic screen and identify insomniac as a modulator of retrograde signaling (after the silencing of the GluRII receptors). Besides its relevance for sleep maintenance, insomniac is essential in the fly muscle to activate the retrograde signaling from the muscle to the motor neuron, acting either downstream or in parallel to CaMKII. Finally, the authors focused on the

molecular function of insomniac as an E3 ubiquitin ligase and showed that insomniac is relevant for the mono-ubiquitination of synaptic proteins.

This is an interesting paper that presents a new player in neuronal plasticity. There are however a couple of problems that I hope the authors can address. These relate to the difficulty in interpretation of the results that are not straightforward due to lacking data, the improper reporting of statistical analyses, and an imprecise description of the methodology.

1. Around 400 genes were chosen to start the genetic screen. However, since several diseases present a defect in the homeostatic control of the synaptic plasticity, I am a bit unsure as to the justification of the choice of FMRP targets to create this collection. Can this be clarified?

The Reviewer is certainly correct that a variety of diseases are associated with defects in homeostatic synaptic plasticity (Ramocki and Zoghbi, 2008; Wondolowski and Dickman, 2013). As detailed in the response to Reviewer 1 Point 7 above, the screen described in Fig. 1 is ultimately a gene discovery approach and was not meant to imply that that only genes targeted by FMRP would be involved in PHP signaling. The specific reasons we chose FMRP targets (which constituted ~40% of all the genes screened) were **1)** the intriguing and specific links between FMRP and homeostatic plasticity at synapses in a variety of systems (Henry, 2011; Lee et al., 2018; Soden and Chen, 2010; Zhang et al., 2018); **2)** The emergence of numerous studies that have defined hundreds of transcripts that encode synaptic proteins targeted for modulation by FMRP (Bassell and Warren, 2008; Muddashetty et al., 2007; Schutt et al., 2009); and **3)** Several studies at the *Drosophila* NMJ that implicate postsynaptic translation to be a process involved in PHP (Kauwe et al., 2016; Penney et al., 2012). We have revised the Results section to make these points more clearly (**lines 96-103**) and thank the Reviewer for these suggestions.

2. The authors show that the loss of insomniac results in a lack of ubiquitination of proteins at the synapse. However, there is no direct link to the altered retrograde signaling observed in the mutants. In line with the previous comments, a broader screen (that contains a larger number of genes) could help to elucidate the downstream events that link insomnia loss to the function of the presynaptic neuron.

We appreciate this point, a sentiment shared by the other two Reviewers. It required over two years for us to complete the screens that successfully identified *inc* presented in Fig. 1. While the time given for the revision of this manuscript was not sufficient to conduct an entirely new large scale genetic screen, we did perform a targeted screen of 23 genes that putatively interact with *Inc* and *Cul3* (detailed in our response to Reviewer 1 Point 3 above). These results are now presented in a new Fig. 6. As indicated above and throughout the revised manuscript, we are very excited about the identification of *Peflin* as a putative *Cul3/Inc* co-adaptor that confers Ca^{2+} regulation in the context of homeostatic signaling at the PSD. Further, we have investigated whether there is a link between *Cul3/Inc* and *Multiplexin*, a key proposed retrograde signal. These results are detailed in our response to Reviewer 2 Point 2 above and included in the new Fig. 6 and throughout the revised manuscript. We thank all 3 Reviewers for these suggestions, which have substantially improved our manuscript.

3. The authors opted for a graphical representation of the results of the screen instead of showing the individual data for each of the genes that can modulate the retrograde signaling (without changing the basal synaptic strength). While this option allows a rapid

visualization of the results, it is important to present the identity of these genes (in supplementary material for example). In addition, it is important to present the data used for the classification of the positive genes in pre- or post-synapse (lines 142-144).

We apologize if this Reviewer did not receive Supplemental Tables 1 and 2, which detailed information of all the genes, mutant alleles, and RNAi lines screened as well as associated data related to the screen shown in Fig. 1. Substantially revised and improved versions of these tables are included in the current version of this manuscript as described above (please see Response to Reviewer 1 Points 7 and 8). We agree this information is important.

In terms of showing additional details of the positive hits from the screen, we have included details on one positive hit from the screen (*fife*) in Supplementary Table 1, as well as information about the genes found to function in baseline transmission, presented in Supplementary Table 2. However, we have elected not to reveal the identity of the five candidate genes we identified to function presynaptically to be necessary for PHP expression for two reasons. First, we are actively working on characterizing these genes and their potential roles in PHP expression, which we anticipate will be the subject of future manuscripts that properly focus on these genes. Of course, the full details, including pre vs. post function, will be presented in these manuscripts. Second, the data on the additional positive hits are still in preliminary stages, and we have not fully validated their roles in PHP expression; additional mutant alleles, rescue lines, and other reagents are necessary to develop. Therefore, we are not prepared, at this point, to reveal the additional positive hits until we have fully validated and characterized their functions in PHP signaling.

4. While the new insomniac mutants (*kk3* and *kk4*) seem to have the same phenotypes as the available insomniac mutants (1 and 2), the creation of the mutants needs to be better controlled. Were the mutants checked for off targets of the CRISPR/Cas9 activity? Did the authors check the insomniac mRNA levels?

In the original manuscript, we defined several new *inc* alleles that are predicted to lead to early stop codons in the *inc* transcript (Fig. 2). This was validated by **1)** generation of a guide RNA that was selected to have no off target effects based on the algorithms described in (Hsu et al., 2013); **2)** genomic sequencing; **3)** detailed characterization of two independently derived mutants (*inc^{kk3}* and *inc^{kk4}*); **4)** Immunoblot analysis showing no Inc protein in either of the new *inc* mutations (null alleles; Fig. 2); **5)** Demonstration of similar defects in sleep behavior as reported in the original *inc¹* and *inc²* alleles (Stavropoulos and Young, 2011); **6)** Tissue-specific rescue of both the sleep and PHP phenotypes in *inc* mutant backgrounds (Fig. 2 and Fig. S1). In addition, after generation of the *inc^{kk3}* and *inc^{kk4}* alleles, we outcrossed these alleles for 8 generations to control for the genetic background. As part of this outcrossing, we ensured new isogenic second and third chromosomes were incorporated that eliminated any chance of off-target effects on these chromosomes (Over 80% of the fly genome). Finally, because extensive recombination occurred on the X chromosome containing the new *inc* alleles during this outcrossing, we are confident that the potential for off-target CRISPR-Cas9 activity on the X was well controlled. Together, these approaches provide strong evidence that possible off target effects of Cas9 activity were highly controlled. We have now added additional details of these approaches to the Methods (**lines 483-486**) and thank the Reviewer for these suggestions.

5. Can the authors comment on the apparent contradiction between the supplementary figure 1 (where the EPSC, mEPSC and quantal content are altered) and the statement on the maintenance of the basal synaptic transmission of *inc* mutants (lines 119-123 and 158-159)?

We apologize if this was confusing. A previous study performed NMJ electrophysiology on *inc¹/inc²* alleles and reported reduced EPSP and quantal content values (Li et al., 2017). We did not observe significant changes in baseline synaptic transmission in the new CRISPR *inc^{kk3}* and *inc^{kk4}* alleles, nor in homozygous *inc¹* alleles (Fig. 2 and Fig. S1). We did confirm the same defect in baseline electrophysiology in the *inc¹/inc²* combination (presented in Fig. S1), which suggests this phenotype is likely due to the unique genetic background and allelic combination in this condition. This is now more fully clarified in the legend of Fig. S1.

6. The methods and figure legends need to be improved. For example:

a. Fig 1S g and h: Are all the animals used females? If so, please confirm that the genotypes of the orange bar is "*inc1/inc1, inc-Gal4*" and the red is "*inc1/inc1, inc-Gal4; UAS-smFP-inc/+*". Also, can the authors justify the difference of the total daily sleep in the wild-type flies in g and h? A value of 1000 minutes of sleep is typical for male flies and not for females.

We apologize if this was unclear. As the Reviewer indicates, sleep is reported for males in Fig. S1g and females in Fig. S1h. This is because of the necessary genetic backgrounds of each experiment, which required we use females for Fig. S1g and males for S1h. We have made this clear in the revised Fig. S1 and in the associated legend.

b: Can the authors clarify the method to quantify the intensity of the fluorescent signal in figure 3? According to the methods section, the signal of a z-stack was projected using the maximum intensity function, which is a non-standard method to quantify those images. Instead, a sum of all the imaged z-planes is used to quantify the fluorescent signal.

We have clarified and added additional details of how intensity was quantified in Fig. 3. We have also performed additional experiments and analyses with further details and explanations in the revised Methods (**lines 561-564**).

c: Please provide the method used to match the mammalian and *Drosophila* homologs.

We have added additional details to the Methods (**lines 447-449**), where we used the NCBI protein Basic Local Alignment Search Tool (BLAST) algorithm (Pearson, 2013), cross referenced with similar analyses on Flybase (www.flybase.org), which assigns *Drosophila* genes to mammalian homologs.

d: Which type of Cas-9 was used? Which method was used to test for off-targets (line 363)?

We used the Cas9 reported in (Gratz et al., 2013) and generated the guide RNA sequence to avoid off target effects using the scoring algorithm detailed in (Hsu et al., 2013). This is now detailed in the revised Methods (**lines 471-474**).

e: Please provide a detailed description of the statistical test to each graph, since it is difficult to understand some of the comparisons. For example, figure 4c, is the ns compared to the 100%? Also, is the * a comparison to the 100% or to the value of the black column? In other words, was it used a one-sample t-test in this case?**

We apologize that some of the statistics were unclear in the legends of the original manuscript. We have revised the legends to include more details of the statistical tests used and what each comparison is referring to. Briefly, we used ANOVA for comparisons between multiple genotypes, and a Student's t test in cases in which only two groups were being compared. For clarity, we only show the significance value for each compared to the relevant control in each data set. For comparisons between groups that not controls, these values are shown in the revised Supplementary Table 3. We have also added more statistical details to the revised Supplementary Table 3.

7. Please provide improved images for the post-synaptic location of the endogenous *inc-smFP*, since the image provided in figure 4a does not clearly show the presence of *insomniac* in that structure.

We agree this is an important point. We have now moved the images of the entire NMJ to a supplemental figure (Supplemental Fig. 4), and provided new images focused on a few boutons specifically labeled with *Inc^{smFP}* (or Flag-Cul3), HRP, and DLG. In addition to providing images of *Inc^{smFP}*, we have added new data and images of *Inc^{smFP}* with RNAi-knock down of *inc* presynaptically (to better highlight postsynaptic *Inc^{smFP}*), and, importantly, Flag-Cul3 expressed exclusively in the postsynaptic muscle. This new data is all included in a substantially revised Fig. 4, which we think very much strengthens the manuscript.

8. What is the link between the lack of mono-ubiquitination at the post-synapse (in the *insomnia* mutants after PhTx) and the decreased retrograde signaling? Also, the authors failed to demonstrate that the levels of ubiquitin are not altered at the synapse (which could be a cause for the decreased protein ubiquitination in the mutants) or to define the proteins that are targets of *insomniac* at the post-synapse that play a role in this mechanism.

We thank the Reviewer for these points. The question of how *Inc* and mono-ubiquitination are related to retrograde homeostatic signaling was shared by Reviewers 1 and 2, and we have responded to these points with new data in the manuscript. This is detailed in our response to Reviewer 1 Point 3 and Reviewer 2 Point 2 above.

The question of whether the levels of Ubiquitin are altered in *inc* mutants at baseline in terms of FK1 staining was detailed in our response to Reviewer 1 Point 6 above. We find no significant difference in the FK1 signal in *inc* mutants compared to controls, which is now included in the revised Fig. 5. In addition, the FK2 significance data in *inc* mutants after PhTx compared to baseline was included in the original manuscript in the Supplemental Table 3 (now Supplemental Table 4 in the revised manuscript). We find no significant difference in the FK2 signal at the NMJ in *inc* mutants compared to its own baseline or to the wild type baseline value. We have included this data now in the revised Fig. 5 and additional details are shown in the revised Supplemental Table 4.

REFERENCES

- Bassell, G.J., and Warren, S.T. (2008). Fragile X syndrome: loss of local mRNA regulation alters synaptic development and function. *Neuron* 60, 201-214.
- Bennett, E.J., Rush, J., Gygi, S.P., and Harper, J.W. (2010). Dynamics of cullin-RING ubiquitin ligase network revealed by systematic quantitative proteomics. *Cell* 143, 951-965.
- Giot, L., Bader, J.S., Brouwer, C., Chaudhuri, A., Kuang, B., Li, Y., Hao, Y.L., Ooi, C.E., Godwin, B., Vitols, E., *et al.* (2003). A protein interaction map of *Drosophila melanogaster*. *Science* 302, 1727-1736.

- Gratz, S.J., Cummings, A.M., Nguyen, J.N., Hamm, D.C., Donohue, L.K., Harrison, M.M., Wildonger, J., and O'Connor-Giles, K.M. (2013). Genome engineering of *Drosophila* with the CRISPR RNA-guided Cas9 nuclease. *Genetics* 194, 1029-1035.
- Haghighi, A.P., McCabe, B.D., Fetter, R.D., Palmer, J.E., Hom, S., and Goodman, C.S. (2003). Retrograde control of synaptic transmission by postsynaptic CaMKII at the *Drosophila* neuromuscular junction. *Neuron* 39, 255-267.
- Hein, M.Y., Hubner, N.C., Poser, I., Cox, J., Nagaraj, N., Toyoda, Y., Gak, I.A., Weisswange, I., Mansfeld, J., Buchholz, F., *et al.* (2015). A human interactome in three quantitative dimensions organized by stoichiometries and abundances. *Cell* 163, 712-723.
- Henry, F.E. (2011). A fragile balance at synapses: new evidence supporting a role for FMRP in homeostatic plasticity. *J Neurosci* 31, 6617-6619.
- Hsu, P.D., Scott, D.A., Weinstein, J.A., Ran, F.A., Konermann, S., Agarwala, V., Li, Y., Fine, E.J., Wu, X., Shalem, O., *et al.* (2013). DNA targeting specificity of RNA-guided Cas9 nucleases. *Nat Biotechnol* 31, 827-832.
- Hudson, A.M., and Cooley, L. (2010). *Drosophila* Kelch functions with Cullin-3 to organize the ring canal actin cytoskeleton. *J Cell Biol* 188, 29-37.
- Hudson, A.M., Mannix, K.M., and Cooley, L. (2015). Actin Cytoskeletal Organization in *Drosophila* Germline Ring Canals Depends on Kelch Function in a Cullin-RING E3 Ligase. *Genetics* 201, 1117-1131.
- Hutchins, J.R., Toyoda, Y., Hegemann, B., Poser, I., Heriche, J.K., Sykora, M.M., Augsburg, M., Hudecz, O., Buschhorn, B.A., Bulkescher, J., *et al.* (2010). Systematic analysis of human protein complexes identifies chromosome segregation proteins. *Science* 328, 593-599.
- Kauwe, G., Tsurudome, K., Penney, J., Mori, M., Gray, L., Calderon, M.R., Elazouzzi, F., Chicoine, N., Sonenberg, N., and Haghighi, A.P. (2016). Acute Fasting Regulates Retrograde Synaptic Enhancement through a 4E-BP-Dependent Mechanism. *Neuron* 92, 1204-1212.
- Koles, K., Yeh, A.R., and Rodal, A.A. (2015). Tissue-specific tagging of endogenous loci in *Drosophila melanogaster*. *Biol Open* 5, 83-89.
- Lee, K.Y., Jewett, K.A., Chung, H.J., and Tsai, N.P. (2018). Loss of Fragile X Protein FMRP Impairs Homeostatic Synaptic Downscaling through Tumor Suppressor p53 and Ubiquitin E3 Ligase Nedd4-2. *Hum Mol Genet*.
- Li, Q., Kellner, D.A., Hatch, H.A.M., Yumita, T., Sanchez, S., Machold, R.P., Frank, C.A., and Stavropoulos, N. (2017). Conserved properties of *Drosophila* Insomniac link sleep regulation and synaptic function. *PLoS Genet* 13, e1006815.
- Li, X., Goel, P., Chen, C., Angajala, V., Chen, X., and Dickman, D.K. (2018). Synapse-specific and compartmentalized expression of presynaptic homeostatic potentiation. *Elife* 7.
- McGourty, C.A., Akopian, D., Walsh, C., Gorur, A., Werner, A., Schekman, R., Bautista, D., and Rape, M. (2016). Regulation of the CUL3 Ubiquitin Ligase by a Calcium-Dependent Co-adaptor. *Cell* 167, 525-538 e514.
- Muddashetty, R.S., Kelic, S., Gross, C., Xu, M., and Bassell, G.J. (2007). Dysregulated metabotropic glutamate receptor-dependent translation of AMPA receptor and postsynaptic density-95 mRNAs at synapses in a mouse model of fragile X syndrome. *J Neurosci* 27, 5338-5348.
- Pearson, W.R. (2013). An introduction to sequence similarity ("homology") searching. *Curr Protoc Bioinformatics Chapter 3*, Unit3 1.
- Penney, J., Tsurudome, K., Liao, E.H., Elazzouzi, F., Livingstone, M., Gonzalez, M., Sonenberg, N., and Haghighi, A.P. (2012). TOR is required for the retrograde regulation of synaptic homeostasis at the *Drosophila* neuromuscular junction. *Neuron* 74, 166-178.
- Ramocki, M.B., and Zoghbi, H.Y. (2008). Failure of neuronal homeostasis results in common neuropsychiatric phenotypes. *Nature* 455, 912-918.

- Rodal, A.A., Blunk, A.D., Akbergenova, Y., Jorquera, R.A., Buhl, L.K., and Littleton, J.T. (2011). A presynaptic endosomal trafficking pathway controls synaptic growth signaling. *J Cell Biol* 193, 201-217.
- Schutt, J., Falley, K., Richter, D., Kreienkamp, H.J., and Kindler, S. (2009). Fragile X mental retardation protein regulates the levels of scaffold proteins and glutamate receptors in postsynaptic densities. *J Biol Chem* 284, 25479-25487.
- Soden, M.E., and Chen, L. (2010). Fragile X protein FMRP is required for homeostatic plasticity and regulation of synaptic strength by retinoic acid. *J Neurosci* 30, 16910-16921.
- Stavropoulos, N., and Young, M.W. (2011). insomniac and Cullin-3 regulate sleep and wakefulness in *Drosophila*. *Neuron* 72, 964-976.
- Wang, J., Huo, K., Ma, L., Tang, L., Li, D., Huang, X., Yuan, Y., Li, C., Wang, W., Guan, W., *et al.* (2011). Toward an understanding of the protein interaction network of the human liver. *Mol Syst Biol* 7, 536.
- Wang, T., Hauswirth, A.G., Tong, A., Dickman, D.K., and Davis, G.W. (2014). Endostatin is a trans-synaptic signal for homeostatic synaptic plasticity. *Neuron* 83, 616-629.
- Wondolowski, J., and Dickman, D. (2013). Emerging links between homeostatic synaptic plasticity and neurological disease. *Front Cell Neurosci* 7, 223.
- Zhang, Z., Marro, S.G., Zhang, Y., Arendt, K.L., Patzke, C., Zhou, B., Fair, T., Yang, N., Sudhof, T.C., Wernig, M., *et al.* (2018). The fragile X mutation impairs homeostatic plasticity in human neurons by blocking synaptic retinoic acid signaling. *Sci Transl Med* 10.

REVIEWERS' COMMENTS:

Reviewer #1 (Remarks to the Author):

The authors have added additional data to further strengthen the role of ins and Cul3 in the postsynaptic compartment. In addition, the new links to Pef and calcium regulation provide new insights into how monoUb might link to calcium signaling during homeostatic plasticity. Overall, it is a very nice study and important work for the field.

Reviewer #2 (Remarks to the Author):

The authors have done an excellent job in addressing all concerns raised during the previous round of review process and in my opinion, the revised manuscript can now be recommended for the publication in the journal as it is.

Reviewer #3 (Remarks to the Author):

I am very satisfied with the author's responses and changes to review, this is a nice paper. There is one point that I think could be improved. In my point 6b I asked for clarifications on the methodology used, and I was not clear enough (apologies for this): When quantifying the level of a protein by immunohistochemistry, the authors use a maximum intensity projection from a z-stack. The best would be to use the sum of the signal intensities.

RESPONSE TO REVIEWER

1. Reviewer #3 states: *I am very satisfied with the author's responses and changes to review, this is a nice paper. There is one point that I think could be improved. In my point 6b I asked for clarifications on the methodology used, and I was not clear enough (apologies for this): When quantifying the level of a protein by immunohistochemistry, the authors use a maximum intensity projection from a z-stack. The best would be to use the sum of the signal intensities.*

We apologize for not fully responding to the Reviewer's point in the previous revision. During the initial phases of this project as well as other projects that rely on performing quantitative measurements of fluorescence intensities, we rigorously assessed several different methods for quantifying the levels of a protein by immunohistochemistry. This included maximum intensity projections, sum of signal intensities (from individual Z-stacks), normalization to internal controls and assessments with fluorescently conjugated beads of known sizes. In the course of this work, which began over four years ago in close collaboration with Nikon Instruments and their analysis and software development team, we found very little difference in the ultimate results when quantifying sum fluorescence on a max intensity projection image vs. summation of signal intensities from each individual Z-stack. We also discussed this methodology with leading experts in our field, most of whom confirmed they use max intensity projections for their studies (e.g. Graf et al., *Neuron*, 2009; Bohme et al., *Nat Commun*, 2019; Gratz et al., *J. Neurosci*, 2019). Finally, we found that in certain conditions, such as cases in which antibodies or tagged constructs had a suboptimal signal-to-noise ratio and/or when background was high, analyzing the sum of signal intensities could lead to artifacts. Since we found the relative changes and ultimate result was not significantly different using the two methodologies, we have settled on using the maximum intensity projection methodology in our studies.

That being said, we did perform the sum of signal intensity analysis on the pCaMKII data in Fig. 3 that the Reviewer originally pointed out. As shown in the Reviewer Figure below, we found no significant difference in the ultimate signal intensity change using either the max intensity or sum of signal intensity methodology. We now added this information to the Methods section (lines 596-598) in the revised manuscript and we thank the Reviewer for raising this point and providing us an opportunity to discuss and justify the quantitative imaging methodology used in this study.

Reviewer Figure: Comparison of pCaMKII signal intensity quantification using both maximum intensity projection and sum of signal intensity methodology. Data presented in Figures 3e and 3f in the manuscript.